# No LLM Solved Yu Tsumura's 554th Problem

## Abstract

We show, contrary to the optimism about LLM's problem-solving abilities, fueled by the recent gold medals at the International Math Olympiad (IMO) that LLMs attained, that a problem exists—Yu Tsumura's 554th problem—that a) is within the scope of an IMO problem in terms of proof sophistication, b) is not a combinatorics problem, which have caused issues for LLMs, c) requires fewer proof techniques than typical hard IMO problems, d) has a publicly available solution (likely in the training data of LLMs), and e) that cannot be readily solved by *any* existing off-the-shelf LLM (commercial or open-source). We include an analysis of the output traces of 16 SOTA LLMs. Additionally, we compare the generic LLM output to a new proof by a former IMO participant, carried out in a small study, which is significantly better motivated than the original, publicly-available proof, and elaborate on the differences in LLM and human *proof quality*.

## 1 Introduction

The results achieved by several commercial companies at the International Mathematical Olympiad[1] in 2025 (IMO25) have been hailed as a milestone in AI in press releases.[2], as well as "awfully impressive" by some researchers [3]. Lending credence to these statements is a recent replication of some of these results in a scientific setting (Huang & Yang, 2025) by using a more complex verification scheme, as well as an the *OpenAI × AIMO* evaluation from March, released in August, on a version of OpenAI o3's model that solved 47/50 Olympiad-level hidden math problems there were used for the AIMO2 competition.[4] Due to the difficulty of the involved problems, these results paint a very optimistic future for the reasoning abilities of state-of-the-art LLMs.

In this paper, we present a counterclaim that offers a more nuanced perspective on the current state of affairs. The previous results show that there exist LLMs that solve math problems that require intricate reasoning abilities, for which likely no solution was in the training data ahead of time, due to the recency of the problems. We show the converse: There exist mathematical problems that require intricate reasoning abilities that no current off-the-shelf LLMs have, *for which a full solution exists online, posted at a time that predates the advent of LLMs*.[5]

For IMO25 problems, both AI systems solving the problems formalized in Lean4 were evaluated, as well as system processing the problem formulated in natural-language. We focus our analysis solely on the natural-language problems.

Specifically, we show that there exists a problem that none of the current set of widely used LLMs, whether proprietary or open-weight, can solve. This problem is publicly available and is Yu Tsumura's 554th problem.[6] It is a group theory problem, but we emphasize that no specialized knowledge of group theory is needed. We reproduce it below:

---

[1]`imo-official.org`

[2]`https://blog.google/products/gemini/gemini-2-5-deep-think/`

[3]`https://www.nature.com/articles/d41586-025-02343-x`

[4]`https://aimoprize.com/updates/2025-09-05-the-gap-is-shrinking`

[5]On this `archive.org` link, the year 2017 is the first time the problem is listed online. This link contains a copy of the statement and the proof.

[6]Yu Tsumura's 554th problem.

> *Let $x$, $y$ be generators of a group $G$ with relations*
> $$xy^2 = y^3x,$$
> $$yx^2 = x^3y.$$
> *Prove that $G$ is the trivial group.*

Its proof, which is also provided in the link above, is short and requires nothing more than clever symbolic manipulation–a task that LLMs solving Olympiad-level math problems need to possess. In fact, this problem is similar to an IMO problem, such as a functional equation problem or inequality, where there are established proof techniques. This should be the easiest case for an LLM in terms of problem difficulty. While the proof, provided in the link above, utilizes the concepts of conjugacy and the order of a group element, these are only mentioned once and can be unpacked. This makes the proof independent of any specific knowledge about group theory.

We speculate that the problem is difficult for LLMs because solving the problem involves a deep search through identities that can be derived from the original relations. There are two potential reasons why this poses a problem:

- The probability of the LLM hallucinating/making an algebraic error before finding the required identities is very high.

- The LLM is not trained to search to a high enough expression depth.

Lastly, we note that 60 members of the public have self-reported on Yu Tsumura's website that they solved the problem. In addition to this, we conducted a $n = 1$ study to highlight the differences between LLM's mathematical reasoning abilities and human mathematical reasoning abilities: Even though LLMs are trained on vastly more data, they were not able to solve the problem, as shown in Section 2; on the other hand, we asked a former IMO participant, who did not have any exposure to group theory, to learn about the basic definitions necessary to understand Yu Tsumura's 554th problem and to attempt to solve the problem. He succeeded, and devised a new proof strategy to solve Yu Tsumura's problem. We comment on his approach in Section 3 which shows that whereas the LLMs seem to need to spend a lot of their time just trying random algebraic manipulations with little clear direction, the IMO participant clearly motivated different proof strategies, which highlights a completely approach to problem-solving, that LLMs lack.

## 2 RESULTS

All our evaluations were performed one-shot, i.e., a single attempt was made to obtain the answer. Our assessment is made from the point of view *of an end user at the present point in time*. Thus, we are assessing whether the model can answer Yu Tsumura's 554th problem *robustly*, which means that the model has to produce the correct answer most of the time, making a one-shot evaluation should be sufficient. Repeated evaluations might produce correct proofs, but if it takes a best-of-$n$ approach, majority voting, or other techniques to elicit them, from the perspective of the end user, this would be a different model that is evaluated (namely, one where the tested LLM incorporates an output refinement strategy on top, that mirrors the repeating-evaluation framework ).

The list of models that we queried is given in Table 2. These models arguably represent the state of the art among publicly available options. Although this list is not exhaustive, these models likely outperform most others and are the most highly rated ones on website such as `lmarena.ai` (except GPT-4.5). Therefore, we reason that if these models are unable to solve the problem, it is unlikely that other comparable or less capable models will succeed either.

The failures in each case are fatal to the proof. In all cases, the model relied on the error we listed to complete its output. None of the models makes really significant progress before such an error derails the model, or in the case of the "argument incomplete" annotation, the model appears to give up and declares success before much meaningful progress is made.

The fact that our result transcends the various types of LLMs indicates:

| LLM | Access | Eval Date | Failure |
| --- | --- | --- | --- |
| o3-pro (B.1) | OpenRouter | 28 Jul | D |
| o3 (B.2) | online GUI | 1 Aug | C, T |
| o4-mini-high (B.3) | online GUI | 1 Aug | T |
| GPT-4o (B.4) | online GUI | 1 Aug | I |
| Gemini 2.5 Pro (B.5) | online GUI | 1 Aug | A |
| DeepSeek R1 (B.6) | online GUI | 1 Aug | U |
| Claude Sonnet 4 (Ext. Thinking) (B.7) | online GUI | 2 Aug | U |
| Claude Opus 4 (Ext. Thinking) (B.8) | OpenRouter | 2 Aug | T |
| Grok 4 0709 (B.9) | LMArena | 2 Aug | U |
| Kimi K2 (B.10) | OpenRouter | 2 Aug | A, I |
| Qwen3 235B A22B Thinking 2507 (B.11) | OpenRouter | 2 Aug | U |
| GLM-4.5 (B.12) | OpenRouter | 2 Aug | A |
| Gemini 2.5 Deep Think (B.13) | online GUI | 3 Aug | A |
| Llama 4 Maverick (B.14) | LMArena | 3 Aug | U |
| DeepSeek v3 0324 (B.15) | OpenRouter | 3 Aug | I |
| QwQ 32B (B.16) | LMArena | 3 Aug | A, U |
| GPT-OSS-120B (B.17) | OpenRouter | 14 Aug | U |
| GPT-5 Thinking (B.18) | online GUI | 16 Aug | D |

Table 1: A table of all 16 evaluated LLMs on Yu Tsumura's 554th problem, together with the dates at which the models were prompted, and links to the full outputs and detailed failure mode descriptions. For both Claude models the "Extended Thinking" option was turned on. Some models are missing size specifications, e.g., DeepSeek R1 as the GUI, that was used to access the model, did not reveal this information about the underlying model, see Appendix A for more information. None of the listed models were able to solve it flawlessly, as outlined by the failure modes (see key below). We refer to Section B for full output traces, and detailed explanation about the (potentially multiple) critical failure modes, and on which lines of the proof they occur.

Key: A = algebra error, C = missed case, D = incompatible definition,
I = argument incomplete, T = inapplicable theorem,
U = unwarranted assumption/claim

- **Lack of high-quality scientific evaluation.** In contrast to final-answer benchmarks and evaluations, such as OlympiadBench (He et al., 2024), for which automatic assessment is possible, there are few benchmarks for assessing proof-based reasoning, due to the high human effort involved. Exceptions are `matharena.ai`, (Petrov et al., 2025) and the earlier GHOSTS benchmark and evaluation (Frieder et al., 2023) problems, which contains a preliminary assessment on older LLMs on 100 problems from the book by A. Engel, "Problem-Solving Strategies" (Engel, 1998). Benchmarks comprising just six problems, such as the evaluation on IMO25 problems, are too small to make an informed assessment about the (mathematical) reasoning abilities of LLMs. The current results emphasize this, contradicting the optimism that the IMO25 inspires.

- **Outcome misalignment.** The goal is to increase the reasoning abilities of LLMs, which can be measured by the number of problems an LLM can solve. Relying on final-answer benchmarks can skew this. Hence, problems where proof assessment is performed are necessary to establish the baseline of reasoning abilities, and the current failure shows that some gaps may still exist towards final-answer benchmarks.

## 3 HUMAN COMPARISON AND A NEW PROOF

Yu Tsumura's 554th problem belongs to the domain of group theory, which is not a domain that is present at problems from the IMO.

Inspection of the original proof reveals that no specific group theory knowledge or group theoretic proof strategies are needed to solve Yu Tsumura's problem, beyond the definition of a group and a generator. Nonetheless, it is unclear whether the difficulty of the problem is within the reach of an IMO-level competitor.

To clarify this, we carried out an $n = 1$ study with a former IMO25 participant, who was not yet exposed to any group theory.

Precise instructions were provided to him that he was not to look up information about group theory, and to receive any information he needed to understand the foundational group theoretic definitions solely form interactions with ChatGPT. We shared the full, unredacted transcript of interactions with ChatGPT, as well as his write-up of the proof.[7]

The fact that an IMO participant was able to solve this problem demonstrated, perhaps unsurprisingly, that Yu Tsumura's 554th problem is well within the reach of IMO-level students.

More interesting is to observe that the student engaged, without being explicitly prompted to so in any way, to devise a proof that comes close to what is known as a "motivated proof" (Pólya, 1949; Morris, 2020). These are proofs where each step made in the proof is made more transparent by providing clear motivation. It was observed that LLMs struggle with devising motivated proofs (Frieder et al., 2024), and the current paper highlights more strongly the distinction in human and LLM proof quality. This both pertains to the thoughtful outline he shared that represents his thinking process, as well as the final proof that resulted from this process.

What is noticeable about the solution by the IMO participant is that after spotting the identity $xy^{2n}x^{-1} = x^{3n}$, which some LLMs are also able to devise, the participant exploits it by picking special values of $n$.

In particular, the participants notices that we can replace a factor of 2 with a factor of 3, at the expense of wrapping things in another $x$ and $x^{-1}$. But then by just focusing on the power of 3 dividing $n$, specifically by assuming $n$ is divisible by a sufficiently high power of 3, the participant can control the use of the identity. The main thing that is different here is that he keeps focusing on powers, whereas both Yu Tsumura's original solution and others known correct proofs prove some identity involving $y^{27}$ and then go back to using the original identities to knock out the rest of the solution (or, in case of LLMs, just "fiddling" until the whole thing collapses).

The IMO participant's solution is much nicer because it shows why 27 is important in the proof and where it comes from - a motivated proof step.

---

[7]`https://anonymous.4open.science/r/yutsumura_solution-0BCD/`

## 4 LIMITATIONS

Goodhart's law, which states that "when a measure becomes a target, it ceases to be a good measure" is pervasive in machine learning: This principle highlights that model creators often optimize the models to score highly on a given benchmark, rather than equipping the models with the skills that are partially captured by that benchmark and that are needed to succeed on that benchmark. In this regard, we expect that, having emphasized the problems commercial LLMs face on Yu Tsumura's 554th problem, models will soon be adapted to solve this issue (we hypothesize that for some state-of-the-art models, techniques as straightforward as improved test-time training will lead to the problem being solved). Yet, we conjecture that even in this case, other problems be found on which LLMs will struggle across the board.

Our evaluation pertained exclusively to models that reasoned and did not use a RAG pipeline – since the solution is publicly available, such an approach would not have assessed the reasoning skills of the evaluated model. In the case of o3-Pro, it was necessary to explicitly prompt the model not to look up the solution online.

Our protocol was to give each model a single attempt at a solution. It is reasonable to assume that multiple attempts, especially with the more expensive models, may result in a more complete solution. Of course, commercial models may already do this internally, using techniques such as majority voting, or more sophisticated variants thereof. We did not follow this approach, because our analysis pertains solely to see whether the experience of an *end user* interacting with these language models can live up to the expectation genereted by the strong performance on IMO25.

We have focused on publicly released, widely deployed models, especially flagship models. We cannot exclude that there are boutique models or models that are not yet publicly deployed that can reliably solve the problem.

The difference in model capabilities might also be explained by differences in how much training on the test task was performed (Dominguez-Olmedo et al., 2024).

Lastly, a mathematical problem with a proof that relies mainly on symbolic manipulation will pose few issues for a symbolic solver tailored to this type of reasoning. In this regard we expect that an LLM that has access to a tool, such as Vampire [8] or some other solver, and can translate the problem into the necessary formalism, will be able to solve it.

## 5 CONCLUSION

We have demonstrated that there exists at least one problem drawn from a similar distribution in terms of human difficulty and solution strategies as IMO problems, on which LLMs have demonstrated very strong performance to date, on which LLMs nonetheless systematically fail. Thus, subject to the constraints mentioned in Section 4, reasoning in LLMS remains brittle.

The fact that LLMs attained gold medals in the IMO and that, further, an unreleased variant of OpenAI's o3 solved 50 Olympiad-level problems in the OpenAI × AIMO eval, would imply that LLMs should be able to solve Yu Tsumura's 554th problem, too, which we showed is accessible to a human with IMO-level preparation.

Yet, the fact that none of the LLMs solved it highlights that LLMs' "thinking" is different from the thinking of a human, and, in particular, that their reasoning ability is not *transitive*: Solving problems of a similar level of difficulty does not guarantee that another, similar such problem can be solved.

We are cognizant that Yu Tsumura's 554th problem will soon be solved by LLMs, in particular, once attention has been drawn to the fact that it is not solvable. Our analysis was carried out over a timeframe of less than three weeks between July and August 2025, and meant to capture a noteworthy snapshot at the time.

---

[8]https://github.com/vprover/vampire

Nonetheless, this doesn't change our overall message, as in that case other problems that should be accessible to LLMs based on their performance on comprehensive benchmarks, will turn out to be not solvable by a wide set of LLMs.

We conclude with concerns we have going forward: In announcements on strong LLM performance, it is not always clear what score was used. Several common options exist to score problems from a benchmark, such as a binary score per problem (correct/incorrect), or a score that takes into account repeated sampling (pass@$n$, first introduced in (Kulal et al., 2019)). Not being fully explicit in how exactly the methodology was set up can make the numbers hard to interpret.

For very long running commercial models, it will become difficult to rule out human intervention behind the scenes as benchmarks are being carried out, especially if the models are only deployed to a very small number of individuals or can only be afforded by very few researchers. This will potentially skew results of evaluations and make scientific evaluation difficult, if not impossible. We note that better evaluation standards are needed to address this issue. Pre-registered evaluation, akin to pre-registered studies, where time constraints are discussed and fixed in advance, are needed to exclude issues like these, by making sure (among other things) that the time alloted for the evaluation is not longer than the average expected runtime of the model.

## REPRODUCIBILITY STATEMENT

Due to the stochastic nature of LLMs, re-generating the outputs is not possible. However, we have included the full output traces of all LLMs in Appendix B.

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

## A    ADDITIONAL LLM INFORMATION

In order to emulate the end user experience, we used various web GUIs to access the models. In some cases, only these were available.

We accessed DeepSeek R1 via the DeepSeek website. We presume this is DeepSeek R1 0528. This is a 685B model, comprising 671B main model weights.[9] It is accessed by turning on the DeepThink option in the DeepSeek web interface.

The Claude Sonnet 4 model was run with Extended Thinking turned on in the web interface.

The Claude Opus 4 model was accessed via OpenRouter and it spent 146.5s reasoning, indicating that the Extended Thinking mode was switched on.

On release day, GPT-5 suffered technical issues, such as returning responses unrelated to the prompt, or crashes. An output for GPT-5 was obtained, however OpenAI changed the available options quickly over the coming days and we decided not to use the release day response and wait until the models stabilised.

A later attempt on Aug 14 returned an empty answer after 14m 54s.

The included output for GPT-5 was obtained on Aug 16 with GPT-5 Thinking. The mathematical error was essentially the same as the one produced by GPT-5 on release day. However, the Aug 16 generation only thought for 2min 7s vs 4min 43s on release day. We don't know if this was due to server load or a change in the model.

## B    OUTPUT TRACES

This appendix contains the unmodified solutions produced by the model, along with the lines at which critical errors are made. We also consider a substantially incomplete proof to be a critical error.

The prompt for all LLMs was identical: *Let $G$ be a group with generators $x$ and $y$ and relations $xy2 = y3x$ and $yx2 = x3y$. Can you prove that $G$ is the trivial group.* For models that do a web search, we add: *Do not perform a web search to solve this problem* to incentivize the model to reason rather than to regurgitate the web-based proof. For each model below, we first list the errors, and then reproduce the generated proof. At least one error was always critical, and we indicate which.

### B.1    O3-PRO

Lines 3, 8 and 12: The definition of the commutator used is $[x, y] = xyx^{-1}y^{-1}$. The definition of conjugation used is $x^y = y^{-1}xy$. But the identity $[x, yz] = [x, y][x, z]^y$ is not valid with these definitions. (Critical)

```
1  Write the two relations in commutator form.
2
3  (1)   $xy^2 = y^3x  \iff x y^2 x^{-1} = y^3  \iff [x , y^2] = y$
4  (2)   $yx^2 = x^3y  \iff y x^2 y^{-1} = x^3  \iff [y , x^2] = x$
5
6  Step 1.  Express y in terms of the basic commutator
7  Put $c = [x , y]$.
8  Because $[x , y^2] = [x , y][x , y]^y (= c c^y)$ we have
9    $y = c c^y$. (3)
10
11 Multiply (3) on the right by $(c^y)^{-1}$.
12 Since $(c^y)^{-1} = y^{-1} c^{-1} y$,    (3) gives
13
14    $y (c^y)^{-1} = c \iff y y^{-1} c^{-1} y = c \iff c^{-1} y =
       ↪  c$,
```

[9]https://huggingface.co/deepseek-ai/DeepSeek-V3

```
15  hence $y = c^2$. (4)
16
17  So $y$ is the square of the commutator $c$ and therefore commutes
    ↪  with $c$.
18
19  Step 2.  Express $x$ in terms of the commutator
20  Put $d = [y , x]$.  From (2) we similarly get
21
22    $x = d d^x$. (5)
23
24  Because $(d^x)^{-1} = x^{-1} d^{-1} x$, the calculation used
    ↪  above gives
25
26    $d^{-1} x = d  \implies x = d^2$. (6)
27
28  Step 3.  Relate d to c
29  But $[y , x] = [x , y]^{-1}, i.e. d = c^{-1}$. (7)
30
31  Combining (4), (6) and (7) we obtain
32
33    $y = c^2 , x = (c^{-1})^2 = c ^{-2}$. (8)
34
35  Step 4.  $xy = 1$
36  Because powers of the same element commute,
37    $xy = c ^{-2} c^2 = 1, so y = x^{-1}$. (9)
38
39  Step 5.  Finish with either defining relation
40  Insert $y = x^{-1}$ in the first original relation:
41
42    $x y^2 = y^3 x \implies x·x ^{-2} = x ^{-3} · x$ ,
43                        i.e. $x ^{-1} = x ^{-2}$.
44
45  Multiplying by $x^2$ gives $x = 1$, and then (9) forces $y = 1$.
46
47  Hence every generator of $G$ is the identity; therefore
48
49    $G$ is the trivial group.
```

## B.2  o3

`Lines 36-37:` The value $k$ need not be an integer if $n$ is infinity. (Critical)

`Lines 36-41:` The given map involves $x$ and thus $y$ need not be sent to an element of $\langle y \rangle$. The model has made the assumption that the automorphism of $G$ (conjugation by $x$) induces an automorphism of the subgroup $\langle y \rangle$. (Critical)

```
1   Below is one way to see that the presentation
2
3   $$
4   G=\langle x,y\mid xy^{2}=y^{3}x,\;yx^{2}=x^{3}y\rangle
5   $$
6
7   defines nothing more than the trivial group.
8
9   ---
10
11  ### 1.  The abelianisation is trivial
12
13  Send $x\mapsto \bar x,\;y\mapsto\bar y$ in the free abelian group
    ↪   on two generators.
14  The two relators become
15
16  $$
17  \bar x+2\bar y=3\bar y+\bar x,\qquad
18  \bar y+2\bar x=3\bar x+\bar y,
19  $$
20
21  and both reduce to $ \bar x=\bar y=0$.
22  Hence $G^{\text{ab}}=G/[G,G]=1$; in particular $G=[G,G]$ is
    ↪   *perfect*.
23  If we can show that $G$ is abelian, it will therefore be trivial.
24
25  ---
26
27  ### 2.  Conjugating the cyclic subgroups
28
29  Write $n=\lvert y\rvert\;(1\le n\le\infty)$.
30  From $xy^{2}=y^{3}x$ we get
31
32  $$
33  x y^{2}x^{-1}=y^{3}. \tag{2.1}
34  $$
35
36  Because the subgroup $\langle y\rangle$ is cyclic, there is an
    ↪   integer
37  $k\;(1\le k\le n)$ such that
38
39  $$
40  x y x^{-1}=y^{k}. \tag{2.2}
41  $$
42
43  Squaring (2.2) and comparing with (2.1) yields the congruence
44
45  $$
46  2k\equiv 3\pmod n, \qquad\gcd(k,n)=1. \tag{2.3}
```

$$

Exactly the same argument with the second relator $yx^{2}=x^{3}y$ gives numbers
$m=\lvert x\rvert$ and $\ell$ satisfying

$$
y x y^{-1}=x^{\ell},\qquad 2\ell\equiv 3\pmod m,\qquad\gcd(\ell,m)=1. \tag{2.4}
$$

*If $n=\infty$* then (2.3) would read $2k=3$ in the integers, impossible.
Thus $y=1$ and, inserting this in the first relator, $x=1$.
So from now on we may assume $n,m<\infty$; necessarily $n,m$ are **odd** (otherwise $2$ would not be invertible mod $n$ or $m$).

---

### 3.  A useful commutator

By (2.2)-(2.4),

$$
[x,y]=x y x^{-1}y^{-1}=y^{k-1}=x^{1-\ell}. \tag{3.1}
$$

Because of the congruences $2k\equiv3$ and $2\ell\equiv3$ we have

$$
\gcd(k-1,n)=\gcd(\ell-1,m)=1. \tag{3.2}
$$

(Indeed, if a divisor $d$ satisfied $d\mid n$ and $d\mid k-1$ then $k\equiv1\pmod d$; reducing $2k\equiv3\pmod n$ modulo $d$ would give $2\equiv3\pmod d$, hence $d=1$.)

From (3.1)-(3.2) the element

$$
c:=y^{k-1}=x^{1-\ell}
$$

has order

$$
\lvert c\rvert=n=\lvert y\rvert = m=\lvert x\rvert,
$$

so $\langle c\rangle=\langle y\rangle=\langle x\rangle$.
Thus **both** generators are powers of the *same* element $c$; consequently $G$ is cyclic and therefore abelian.

---

```
### 4.  Finish the argument

We have shown

* $G$ is abelian (Step 3),
* $G$ is perfect (Step 1).

The only perfect abelian group is the trivial one, so $G=1$.
Equivalently $x=1$ and $y=1$.

$$
\boxed{\;G\text{ is the trivial group.}\;}
$$

---

### Why the key step works

The heart of the proof is the observation that the first relator
↪  lets us
*measure* how conjugation by $x$ acts on the cyclic subgroup
↪  generated by $y$; it must multiply exponents by $k$ with
↪  $2k\equiv3$.
Because $k-1$ is coprime to $|y|$, the commutator $[x,y]=y^{k-1}$
↪  still has full order and ends up generating the **same**
↪  cyclic subgroup as $y$.
The second relator yields the mirror-image statement for $x$.
That forces the whole group to be cyclic–hence, after the
↪  abelianisation calculation, trivial.
```

### B.3 O4 MINI HIGH

Lines 21–37: It is shown that $\langle y^2\rangle$ and $\langle y^3\rangle$ are conjugate in $G$. But the following comments require that they be conjugate in $\langle y\rangle$, which is not demonstrated. (Critical)

Lines 53–71: Again, conjugation is in $G$, not in the cyclic subgroup generated by $y$. Thus the stated conclusions are not supported. (Critical)

Lines 93–100: The given map involves $x$ and may not even be a map from $\langle y\rangle$ to itself, let alone an automorphism. (Critical)

```
 1  Here is a self-contained proof that
 2
 3  $$
 4  G = \langle x,y \mid xy^2 = y^3x,\; yx^2 = x^3y\rangle
 5  $$
 6
 7  is the trivial group.
 8
 9  ---
10
11  ## 1. No element can have infinite order
12
13  From
14
15  $$
16  xy^2 = y^3x
17  \quad\Longrightarrow\quad
18  x\,y^2\,x^{-1} = y^3
19  $$
20
21  we see that the subgroups
22
23  $$
24  \langle y^2\rangle
25  \quad\text{and}\quad
26  \langle y^3\rangle
27  $$
28
29  are conjugate in $G$.  But in an infinite cyclic group $\langle
    ↪  y\rangle$, the index of $\langle y^k\rangle$ is $|k|$.
    ↪  Conjugate subgroups have the same index, so you would get
30
31  $$
32  [\<y\>:\<y^2\>] \;=\; 2,
33  \qquad
34  [\<y\>:\<y^3\>] \;=\; 3,
35  $$
36
37  a contradiction.  Hence **$y$ must have finite order**.  By an
    ↪  entirely symmetric argument from
38
39  $$
40  yx^2 = x^3y
41  \quad\Longrightarrow\quad
42  y\,x^2\,y^{-1} = x^3,
43  $$
```

also **$x$ has finite order**.

---

## 2. Reduction to a cyclic, perfect group

Let

$$
m = \mathrm{ord}(y),\quad n = \mathrm{ord}(x).
$$

Because $xy^2x^{-1}=y^3$, conjugation by $x$ carries $\langle y^2\rangle$ onto $\langle y^3\rangle$. In a cyclic group of order $m$, the index of $\langle y^k\rangle$ is

$$
\bigl[\<y\>:\<y^k\>\bigr]
=\frac{m}{\gcd(m,k)}.
$$

Equating the two indices gives

$$
\frac{m}{\gcd(m,2)} \;=\;\frac{m}{\gcd(m,3)}
\;\Longrightarrow\;
\gcd(m,2)=\gcd(m,3).
$$

Since $\gcd(m,2)\in\{1,2\}$ and $\gcd(m,3)\in\{1,3\}$, the only way they are equal is

$$
\gcd(m,2)=\gcd(m,3)=1,
$$

i.e.\ $m$ is coprime to $6$. In particular, **2 and 3 are invertible in $\mathbb{Z}/m\mathbb{Z}$.**

By exactly the same reasoning from the second relation one sees that

$$
\gcd(n,2)=\gcd(n,3)=1
$$

and so **2 and 3 are invertible mod $n$.**

---

## 3. Linearising the conjugations

Because $\gcd(2,m)=1$, the map

$$

```
96   \varphi\colon\<y\>\to\<y\>,\quad
97   \varphi(y^k)=x\,y^k\,x^{-1}
98   $$
99
100  is an automorphism of the finite cyclic group of order $m$.  From
101
102  $$
103  \varphi(y^2)=y^3
104  \quad\Longrightarrow\quad
105  \varphi(y)^2 = y^3,
106  $$
107
108  we deduce that
109
110  $$
111  \varphi(y)=y^c,
112  $$
113
114  where $c$ is the unique integer mod $m$ satisfying
115
116  $$
117  2c\equiv3\pmod m.
118  $$
119
120  Since $\gcd(2,m)=1$, we have
121
122  $$
123  c-1 \;\equiv\;2^{-1}\;\;(\bmod\,m),
124  $$
125
126  so $\gcd(c-1,m)=1$.  Thus
127
128  $$
129  x\,y\,x^{-1} \;=\; y^c.
130  $$
131
132  Similarly, from $\,y\,x^2\,y^{-1}=x^3$ and $\gcd(2,n)=1$ one
     ↪   shows
133
134  $$
135  y\,x\,y^{-1} \;=\; x^d,
136  $$
137
138  where $2d\equiv3\pmod n$ and hence $\gcd(d-1,n)=1$.
139
140  ---
141
142  ## 4. The commutator forces cyclicity {&} perfection
143
144  Compute the two "commutators":
145
146  $$
147  [x,y] \;=\; x\,y\,x^{-1}\,y^{-1}
148  \;=\; y^c\,y^{-1} = y^{\,c-1},
149  $$
150
```

```
$$
[y,x] = y\,x\,y^{-1}\,x^{-1}
= x^d\,x^{-1} = x^{\,d-1}.
$$

But always $[y,x]=[x,y]^{-1}$, so

$$
x^{\,d-1} \;=\; \bigl[y,x\bigr]
\;=\;\bigl[x,y\bigr]^{-1}
\;=\;y^{\,1-c}.
\tag{*}
$$

Now:

1. Since $\gcd(c-1,m)=1$, there is an integer $u$ with

$$
u\,(c-1)\equiv1\pmod m.
$$

Raising $[x,y]=y^{c-1}$ to the $u$th power gives

$$
[x,y]^u \;=\; y^{\,u(c-1)} = y,
$$

i.e.\\

$$
y \;=\; [x,y]^u
\;\in\; [G,G].
\tag{1}
$$

2. Since $\gcd(d-1,n)=1$, there is an integer $v$ with

$$
v\,(d-1)\equiv1\pmod n.
$$

Raising equation $(*)$ to the $v$th power gives

$$
x^{\,v(d-1)} = y^{\,v(1-c)},
$$

but $v(d-1)\equiv1\pmod n$ so $x=y^{\,v(1-c)}$.  Hence

$$
x \;\in\;\langle y\rangle.
\tag{2}
$$
```

```
206  From $(1)$ and $(2)$, **both** generators $x,y$ lie in the
     ↪  *derived subgroup* $[G,G]$, so
207
208  $$
209  [G,G] = G,
210  $$
211
212  i.e.\ $G$ is **perfect**.  On the other hand $(2)$ shows $G$ is
     ↪  cyclic (generated by $y$), hence **abelian**.  The only group
     ↪  which is *both* abelian and perfect is the trivial group.
213
214  ---
215
216  ### Conclusion
217
218  No non-trivial group can satisfy simultaneously
219
220  $$
221  xy^2=y^3x
222  \quad\text{and}\quad
223  yx^2=x^3y,
224  $$
225
226  so
227
228  $$
229  \boxed{G\text{ is the trivial group.}}
230  $$
231
```

### B.4 GPT-4o

`Line 118`: The preceding lines don't demonstrate that conjugation of $y^n$ by $x$ increases the exponent by 1. (Incorrect but not used)

`Line 348`: It is not clear which equation $x = yxy$ comes from. (Incorrect but not used)

`Line 354`: Having made no progress whatsoever, the model concludes the "only consistent solution" is the trivial group. (Critical)

```
1  We are given a group $G$ with generators $x$ and $y$, and two
   ↪   relations:
2
3  $$
4  (1) \quad xy^2 = y^3x, \quad \text{and} \quad (2) \quad yx^2 =
   ↪   x^3y.
5  $$
6
7  Our goal is to **prove that $G$ is trivial**, i.e., $x = y = 1$.
8
9  ---
10
11 ### Step 1: Rewrite the Relations
12
13 Let's manipulate each relation to bring like terms together.
14
15 #### From (1): $xy^2 = y^3x$
16
17 Multiply both sides on the right by $x^{-1}$:
18
19 $$
20 xy^2x^{-1} = y^3.
21 $$
22
23 So,
24
25 $$
26 x y^2 x^{-1} = y^3. \quad \text{(1')}
27 $$
28
29 This says that conjugating $y^2$ by $x$ gives $y^3$, or
30
31 $$
32 x y^2 x^{-1} = y^3 \quad \Rightarrow \quad x y^2 x^{-1} y^{-3} =
   ↪   1.
33 $$
34
35 This is equivalent to:
36
37 $$
38 x y^2 x^{-1} y^{-3} = 1.
39 $$
40
41 #### From (2): $yx^2 = x^3y$
42
43 Multiply both sides on the right by $y^{-1}$:
44
```

$$
yx^2 y^{-1} = x^3.
$$

So,

$$
y x^2 y^{-1} = x^3. \quad \text{(2')}
$$

Similarly,

$$
y x^2 y^{-1} x^{-3} = 1.
$$

---

### Step 2: Define Conjugation Maps

Let's define the following:

* From (1'): $x y^2 x^{-1} = y^3 \Rightarrow x y^2 x^{-1} y^{-3} = 1$
* From (2'): $y x^2 y^{-1} = x^3 \Rightarrow y x^2 y^{-1} x^{-3} = 1$

These say that:

$$
x y^2 x^{-1} y^{-3} = 1 \quad \text{and} \quad y x^2 y^{-1} x^{-3} = 1.
$$

These are conjugation identities showing how $x$ and $y$ act on each other.

Let's now take (1') and apply the same conjugation again.

---

### Step 3: Conjugate (1') Again

From (1'):

$$
x y^2 x^{-1} = y^3.
$$

Now conjugate both sides by $x$:

Left-hand side:

$$
x (x y^2 x^{-1}) x^{-1} = x^2 y^2 x^{-2}.
$$

Right-hand side:

$$
x y^3 x^{-1}.
$$

So,

$$
x^2 y^2 x^{-2} = x y^3 x^{-1}.
$$

But from (1') again:

$$
x y^2 x^{-1} = y^3 \Rightarrow x y^3 x^{-1} = x (x y^2 x^{-1})
x^{-1} = x^2 y^2 x^{-2}.
$$

So this checks out.

But this also tells us that each time we conjugate $y^n$ by $x$,
the exponent increases by one.

So maybe try to **write $x$ and $y$ in terms of each other**.

---

### Step 4: Solve for One Variable

Go back to:

$$
x y^2 = y^3 x.
$$

Let's try to **move all $x$'s to one side**.

Start with:

$$
x y^2 = y^3 x
$$

Multiply both sides on the left by $x^{-1}$, then on the right by
$x^{-1}$:

$$
y^2 = x^{-1} y^3 x \Rightarrow x y^2 x^{-1} = y^3.
$$

As before.

Similarly, from (2):

```
$$
yx^2 = x^3 y \Rightarrow y x^2 y^{-1} = x^3.
$$

Now, write these as:

* $x y^2 x^{-1} = y^3$
* $y x^2 y^{-1} = x^3$

Let's now define $a = x$, $b = y$, to simplify notation.

So:

* $a b^2 a^{-1} = b^3$   (1)
* $b a^2 b^{-1} = a^3$   (2)

Let's try to eliminate one of the variables using these
↪  relations.

---

### Step 5: Try to Find Orders of $x$, $y$

Suppose $x = 1$, the identity.

Then from (1):

$$
1 \cdot y^2 = y^3 \Rightarrow y^2 = y^3 \Rightarrow y^{-1} = 1
↪  \Rightarrow y = 1.
$$

So $x = 1 \Rightarrow y = 1$.

Similarly, if $y = 1$:

From (2):

$$
1 \cdot x^2 = x^3 \Rightarrow x^2 = x^3 \Rightarrow x^{-1} = 1
↪  \Rightarrow x = 1.
$$

So $y = 1 \Rightarrow x = 1$.

So **if either generator is identity, the other must be too**.

Now suppose neither is the identity, but we try to find an order.

Suppose $x$ and $y$ have finite order.

Let's say $x^n = 1$ and $y^m = 1$.

Try to find a contradiction.
```

But a better approach is to **define a new variable** and work
↩  through simplification.

---

### Step 6: Try to Simplify with Substitutions

From the conjugation relation:

$$
x y^2 x^{-1} = y^3 \Rightarrow x y^2 = y^3 x.
$$

Let's isolate $x$. Multiply both sides on the left by $y^{-3}$:

$$
y^{-3} x y^2 = x.
$$

So:

$$
x = y^{-3} x y^2.
$$

Now cancel $x$ from both sides:

Multiply both sides on the left by $x^{-1}$:

$$
1 = x^{-1} y^{-3} x y^2.
$$

Rewriting:

$$
x^{-1} y^{-3} x y^2 = 1.
$$

Similarly, from the second relation:

$$
y x^2 = x^3 y \Rightarrow x^2 = y^{-1} x^3 y \Rightarrow 1 =
x^{-2} y^{-1} x^3 y.
$$

So:

$$
x^{-2} y^{-1} x^3 y = 1.
$$

We now have:

* $x^{-1} y^{-3} x y^2 = 1$   (A)
* $x^{-2} y^{-1} x^3 y = 1$   (B)

Let's work with (A):

$$
x^{-1} y^{-3} x y^2 = 1 \Rightarrow y^{-3} x y^2 = x.
$$

Let's multiply both sides by $x^{-1}$ on the right:

$$
y^{-3} x y^2 x^{-1} = 1.
$$

So:

$$
y^{-3} x y^2 x^{-1} = 1.
$$

That is:

$$
x y^2 x^{-1} = y^3.
$$

We've gone in circles.

Let's take a step back.

---

### Step 7: Try a Free Group Homomorphism

Suppose $G$ is a group with these relations:

$$
G = \langle x, y \mid x y^2 = y^3 x,\ y x^2 = x^3 y \rangle.
$$

Let's define a homomorphism $\phi: G \to \mathbb{Z}$ to abelianize the group.

In the abelianization, $xy = yx$, so we treat all elements as commuting.

From the relations:

1. $x y^2 = y^3 x \Rightarrow x + 2y = 3y + x \Rightarrow 2y = 3y \Rightarrow y = 0$
2. $y x^2 = x^3 y \Rightarrow y + 2x = 3x + y \Rightarrow 2x = 3x \Rightarrow x = 0$

So in the abelianization, $x = y = 0$, so both are trivial in $G^{ab}$.

Therefore, $G$ is a **perfect group** (equal to its commutator subgroup).

Since the abelianization is trivial, we suspect $G$ is trivial or very small.

---

### Step 8: Try the Identity Directly

From:

$$
x y^2 = y^3 x
\Rightarrow x y^2 x^{-1} = y^3
\Rightarrow x y^2 x^{-1} y^{-3} = 1.
$$

Let's define:

$$
r = x y^2 x^{-1} y^{-3}, \quad s = y x^2 y^{-1} x^{-3}.
$$

Then $G = \langle x, y \mid r = 1,\ s = 1 \rangle$.

Now define a **map from $G$ to the trivial group**, mapping $x \mapsto 1, y \mapsto 1$. Clearly both relations are satisfied. So the trivial group is a quotient of $G$.

But we now show that **$x = y = 1$** in $G$.

From:

$$
x y^2 = y^3 x \Rightarrow x y^2 x^{-1} = y^3.
$$

Let's suppose $y \ne 1$, and see that the order of $y$ must be finite.

Let's try small orders.

Suppose $y^n = 1$.

Try $n = 1$: $y = 1 \Rightarrow x = 1$.

Try $n = 2$: then $y^2 = 1 \Rightarrow x = y x y \Rightarrow x = x$, okay.

Try $n = 3$: $y^3 = 1 \Rightarrow x y^2 = x \Rightarrow y^2 = 1 \Rightarrow y = 1$. Contradiction.

So we always get back to $y = 1 \Rightarrow x = 1$.

Therefore, the only consistent solution is:

```
$$
x = y = 1.
$$

---

###  Final Conclusion:

$$
\boxed{G \text{ is the trivial group.}}
$$

All elements reduce to the identity under the given relations.
```

## B.5 GEMINI 2.5 PRO

`Lines 39-40`: $a$ cannot be cancelled from the right, and even if this is a typo, the claimed identity $a = b^3ab^{-1}$ does not follow from $bab = b^4ab^{-1}$. This is an algebra error. (Critical)

```
1   Here is a proof that the group $G$ with generators $x$ and $y$
    ↪  and relations $xy^2 = y^3x$ and $yx^2 = x^3y$ must be the
    ↪  trivial group.
2
3   ### **1. Rewriting the Relations**
4
5   The given relations are:
6   1.   $xy^2 = y^3x$
7   2.   $yx^2 = x^3y$
8
9   To simplify the manipulation of these relations, we can introduce
    ↪  new generators. Let $a = x$ and $b = y^{-1}$. Then $y =
    ↪  b^{-1}$. We can rewrite the original relations in terms of
    ↪  $a$ and $b$.
10
11  * **Relation 1:** $xy^2 = y^3x$ becomes $ab^{-2} = b^{-3}a$.
12  * **Relation 2:** $yx^2 = x^3y$ becomes $b^{-1}a^2 = a^3b^{-1}$.
13
14  Now, we can rearrange these new relations to get expressions for
    ↪  $a$ and $a^2$:
15  * From $ab^{-2} = b^{-3}a$, if we multiply by $b^3$ on the left,
    ↪  we get $b^3ab^{-2} = a$. If we then multiply by $b^2$ on the
    ↪  right, we obtain:
16      $b^3a = ab^2$ (R1')
17  * From $b^{-1}a^2 = a^3b^{-1}$, if we multiply by $b$ on the
    ↪  left, we get $a^2 = ba^3b^{-1}$. If we then multiply by $b$
    ↪  on the right, we obtain:
18      $a^2b = ba^3$ (R2')
19
20  So, the original relations are equivalent to the pair of
    ↪  relations:
21  * $ab^2 = b^3a$
22  * $a^2b = ba^3$
23
24  ---
25  ### **2. Deriving a Key Identity**
26
27  Now, we will use these relations to show that $b$ must be the
    ↪  identity element.
28
29  From $ab^2 = b^3a$, we can derive expressions by progressively
    ↪  multiplying by $b^{-1}$ on the right:
30  * $a = b^3ab^{-2}$
31  * $ab = b^3ab^{-1}$
32  * $ab^2 = b^3a$ (This is R1')
33
34  Let's focus on the identity $ab = b^3ab^{-1}$. We can use this to
    ↪  analyze the expression $bab$:
35  $bab = b(ab)$
36  Substitute the expression for $ab$:
37  $bab = b(b^3ab^{-1}) = b^4ab^{-1}$
```

```
So, we have found that $bab = b^4ab^{-1}$. By canceling $b$ from
↪   the left and $a$ from the right (if $a$ is not the identity),
↪   we get:
$a = b^3ab^{-1}$

Now we have two expressions for $a$:
1.   $a = b^3ab^{-2}$ (from R1')
2.   $a = b^3ab^{-1}$ (derived above)

Equating these two expressions gives:
$b^3ab^{-2} = b^3ab^{-1}$

By canceling $b^3a$ from the left, we get:
$b^{-2} = b^{-1}$

Multiplying by $b^2$ on the right gives:
$e = b$

---
### **3. Proving Triviality**

We have shown that $b = e$. Since we defined $b = y^{-1}$, this
↪   means $y^{-1}=e$, which implies **$y=e$**.

Now we can substitute $y=e$ back into one of the original
↪   relations. Let's use the second relation:
$yx^2 = x^3y$
Substituting $y=e$:
$ex^2 = x^3e$
$x^2 = x^3$

By canceling $x^2$ from the left, we find that **$x=e$**.

Since both generators $x$ and $y$ are the identity element ($e$),
↪   the group $G$ can only contain the identity element.
↪   Therefore, **$G$ is the trivial group**.
```

### B.6 DEEPSEEK R1

Lines 11-13: It is stated that $xyx^{-1} = y^3$. This is claimed to follow from $gyg^{-1} = y^3$. But this is not proved. The claimed proof only shows that $(xyx^{-1})^2 = y^3$. (Critical)

```
1  The group \(G\) is defined by generators \(x\) and \(y\) and
   ↪  relations \(xy^2 = y^3x\) and \(yx^2 = x^3y\). The goal is to
   ↪  show that \(G\) is trivial, meaning \(x = 1\) and \(y = 1\),
   ↪  where 1 is the identity element.
2
3  From the given relations, it is possible to derive additional
   ↪  relations:
4  - From \(xy^2 = y^3x\), conjugation by \(x\) gives \(x y^2 x^{-1}
   ↪  = y^3\). Since conjugation is an automorphism, \(x y^2 x^{-1}
   ↪  = (x y x^{-1})^2\), so \((x y x^{-1})^2 = y^3\).
5  - Similarly, from \(yx^2 = x^3y\), conjugation by \(y\) gives \(y
   ↪  x^2 y^{-1} = x^3\), so \((y x y^{-1})^2 = x^3\).
6
7  Consider the element \(g = x y\). Conjugating \(y\) by \(g\):
8  \[
9  g y g^{-1} = (x y) y (x y)^{-1} = (x y) y (y^{-1} x^{-1}) = x y
   ↪  x^{-1}.
10 \]
11 From the earlier derivation, \(x y x^{-1} = k\) where \(k^2 =
   ↪  y^3\), but also from the relation \(g y g^{-1} = y^3\) (since
   ↪  \(g y g^{-1} = x y x^{-1}\) and \((x y x^{-1})^2 = y^3\)
   ↪  implies that conjugating again or directly, it follows that
   ↪  \(x y x^{-1} = y^3\)). Thus:
12 \[
13 g y g^{-1} = x y x^{-1} = y^3.
14 \]
15 Therefore:
16 \[
17 x y x^{-1} = y^3. \quad (1)
18 \]
19 Similarly, for the other relation, consider \(h = y x\).
   ↪  Conjugating \(x\) by \(h\):
20 \[
21 h x h^{-1} = (y x) x (y x)^{-1} = (y x) x (x^{-1} y^{-1}) = y x
   ↪  y^{-1}.
22 \]
23 From the derivation, \(y x y^{-1} = m\) where \(m^2 = x^3\), and
   ↪  similarly, it follows that:
24 \[
25 h x h^{-1} = y x y^{-1} = x^3.
26 \]
27 Thus:
28 \[
29 y x y^{-1} = x^3. \quad (2)
30 \]
31
32 From equations (1) and (2):
33 - Equation (1): \(x y x^{-1} = y^3\) implies \(x y = y^3 x\).
   ↪  \quad (3)
34 - Equation (2): \(y x y^{-1} = x^3\) implies \(y x = x^3 y\).
   ↪  \quad (4)
```

```
35
36  Now, use the original relations. From the original relation
    ↪   \(xy^2 = y^3x\) \quad (5), and from equation (3), multiply
    ↪   both sides by \(y\):
37  \[
38  x y^2 = y^3 x y. \quad (6)
39  \]
40  Equating (5) and (6):
41  \[
42  y^3 x = y^3 x y.
43  \]
44  Since the group operation is cancellative, left-multiply both
    ↪   sides by \(y^{-3}\):
45  \[
46  x = x y,
47  \]
48  which implies:
49  \[
50  x y x^{-1} = x x^{-1} \implies y = 1. \quad (7)
51  \]
52
53  Similarly, from the original relation \(yx^2 = x^3y\) \quad (8),
    ↪   and from equation (4), multiply both sides by \(x\):
54  \[
55  y x^2 = x^3 y x. \quad (9)
56  \]
57  Equating (8) and (9):
58  \[
59  x^3 y = x^3 y x.
60  \]
61  Left-multiply both sides by \(x^{-3}\):
62  \[
63  y = y x,
64  \]
65  which implies:
66  \[
67  y x y^{-1} = y y^{-1} \implies x = 1. \quad (10)
68  \]
69
70  From (7) and (10), \(x = 1\) and \(y = 1\). Therefore, the group
    ↪   \(G\) is trivial.
71
72  \boxed{\text{The group } G \text{ is trivial.}}
```

### B.7 CLAUDE SONNET 4 (EXTENDED THINKING)

Line 39: The model promises to verify the assumption $y^2 = e$, but this is never independently verified. (Critical)

Lines 51-54: The model assumes $x^2 = e$ without justification. (Critical)

```
Looking at this group presentation, I'll work systematically with
 ↪  the given relations to show that both generators must equal
 ↪  the identity.

Given relations:
1) $xy^2 = y^3x$
2) $yx^2 = x^3y$

**Step 1: Finding the orders of x and y**

From relation (1): $xy^2 = y^3x$

Let me multiply both sides on the left by $y^{-3}$ and on the
 ↪  right by $y^{-2}$:
$$y^{-3}xy^2y^{-2} = y^{-3}y^3xy^{-2}$$
$$y^{-3}x = xy^{-2}$$

Multiplying both sides on the right by $y^2$:
$$y^{-3}xy^2 = x$$

But from our original relation, $xy^2 = y^3x$, so:
$$y^{-3}(y^3x) = x$$
$$x = x$$

Let me try a different approach. From $xy^2 = y^3x$, I'll
 ↪  multiply on the left by $y^{-3}$:
$$y^{-3}xy^2 = y^{-3}y^3x = x$$

So $y^{-3}xy^2 = x$, which means $y^{-3}x = xy^{-2}$.

Multiplying both sides on the right by $y^2$:
$$y^{-3}xy^2 = xy^{-2}y^2 = x$$

From the original relation $xy^2 = y^3x$, we have $y^{-3}xy^2 =
 ↪  y^{-3}y^3x = x$.

This confirms our manipulation. Now, from $y^{-3}x = xy^{-2}$,
 ↪  multiplying on the right by $y^2$:
$$y^{-3}xy^2 = x$$

And since $xy^2 = y^3x$:
$$y^{-3}y^3x = x$$
$$x = x$$

But more importantly, from $y^{-3}x = xy^{-2}$, if we set $y^2 =
 ↪  e$ (which we'll verify), then:
$$y^{-3}x = x$$
$$y^{-3} = e$$
$$y^3 = e$$
```

```
44  **Step 2: Using symmetry for x**

46  Similarly, from relation (2): $yx^2 = x^3y$

48  Following the same process, multiply on the left by $x^{-3}$:
49  $$x^{-3}yx^2 = x^{-3}x^3y = y$$

51  This gives us $x^{-3}yx^2 = y$, so $x^{-3}y = yx^{-2}$.

53  Following similar logic, we get:
54  $$x^3 = e$$

56  **Step 3: Deriving stronger conditions**

58  Now I have $x^3 = e$ and $y^3 = e$.

60  Substituting back into the original relations:

62  From $xy^2 = y^3x$ with $y^3 = e$:
63  $$xy^2 = ex = x$$
64  $$y^2 = e$$

66  From $yx^2 = x^3y$ with $x^3 = e$:
67  $$yx^2 = ey = y$$
68  $$x^2 = e$$

70  **Step 4: Conclusion**

72  Now I have:
73  - $x^2 = e$ and $x^3 = e$
74  - $y^2 = e$ and $y^3 = e$

76  Since $\gcd(2,3) = 1$, there exist integers $a,b$ such that $2a +
    ↪   3b = 1$.

78  For $x$: $x = x^1 = x^{2a + 3b} = (x^2)^a(x^3)^b = e^a \cdot e^b
    ↪   = e$

80  For $y$: $y = y^1 = y^{2a + 3b} = (y^2)^a(y^3)^b = e^a \cdot e^b
    ↪   = e$

82  Therefore, both generators equal the identity, which means $G$ is
    ↪   the trivial group.
```

## B.8 CLAUDE OPUS 4 (EXTENDED THINKING)

Line 20: This is an unwarranted assumption. The conjugation map is a map on $G$ and so the result of conjugating $y$ need not be a power of $y$. (Critical)

```
Looking at the given relations:
- $xy^2 = y^3x$
- $yx^2 = x^3y$

I'll analyze what these relations tell us about the group
↪ structure.

From the first relation $xy^2 = y^3x$, I can rewrite this as:
$$xy^2x^{-1} = y^3$$

This means that conjugating $y^2$ by $x$ gives us $y^3$.

Similarly, from the second relation $yx^2 = x^3y$:
$$yx^2y^{-1} = x^3$$

This means that conjugating $x^2$ by $y$ gives us $x^3$.

Let me define the conjugation map $\phi_x(g) = xgx^{-1}$. Since
↪ conjugation is a homomorphism, and $\phi_x(y^2) = y^3$, I
↪ have:
$$\phi_x(y)^2 = y^3$$

If I let $\phi_x(y) = y^a$ for some integer $a$, then:
$$(y^a)^2 = y^3$$
$$y^{2a} = y^3$$

This means $y^{2a-3} = 1$.

For this to hold, either:
1. $y$ has infinite order, which would require $2a = 3$ in
↪ $\mathbb{Z}$ (impossible for integer $a$)
2. $y$ has finite order $m$, and $2a \equiv 3 \pmod{m}$

So $y$ must have finite order $m$. For the congruence $2a \equiv
↪ 3 \pmod{m}$ to have a solution, we need $\gcd(2,m) | 3$.
↪ Since 3 is odd, we need $\gcd(2,m) = 1$, which means $m$ must
↪ be odd.

By the same reasoning applied to the second relation, $x$ must
↪ have finite odd order $n$, and there exists $b$ such that
↪ $yxy^{-1} = x^b$ where $2b \equiv 3 \pmod{n}$.

Now I'll use both relations together. From our conjugation
↪ equations:
- $xyx^{-1} = y^a$ where $2a \equiv 3 \pmod{m}$
- $yxy^{-1} = x^b$ where $2b \equiv 3 \pmod{n}$

Let me compute $xyxy^{-1}x^{-1}$ in two ways:

First way: $xyxy^{-1}x^{-1} = x(yxy^{-1})x^{-1} = xx^bx^{-1} =
↪ x^b$
```

Second way: $xyxy^{-1}x^{-1} = (xyx^{-1})(yy^{-1}) = y^a \cdot 1$
$\hookrightarrow$  $= y^a$

Therefore: $x^b = y^a$

This is a key relationship. Now, since $x^b = y^a$:
- Applying $\phi_y$: $y(x^b)y^{-1} = y(y^a)y^{-1}$, which gives
$\hookrightarrow$  $(x^b)^b = y^a$, so $x^{b^2} = y^a$
- But also $x^{b^2} = (x^b)^b = (y^a)^b = y^{ab}$

So $y^a = y^{ab}$, which means $y^{a(b-1)} = 1$.

Since $y$ has order $m$, we need $m \mid a(b-1)$.

But we also know:
- From $x^b = y^a$, raising to the $a$-th power: $x^{ab} =$
$\hookrightarrow$  $y^{a^2}$
- Applying $\phi_y$ to this: $y(x^{ab})y^{-1} = y(y^{a^2})y^{-1}$
- This gives $(x^b)^{ab} = y^{a^2}$, so $x^{ab^2} = y^{a^2}$
- But we already had $x^{ab} = y^{a^2}$

So $x^{ab} = x^{ab^2}$, which means $x^{ab(b-1)} = 1$.

Since $x$ has order $n$, we need $n \mid ab(b-1)$.

Now, since $m$ and $n$ are odd, and $a$ and $b$ are odd (as they
$\hookrightarrow$  satisfy $2a \equiv 3 \pmod{m}$ and $2b \equiv 3 \pmod{n}$
$\hookrightarrow$  with odd $m,n$), we have:
- $\gcd(m, ab) = 1$ (since $m$ is odd and $a,b$ are odd)
- Therefore $m \mid (b-1)$ (since $b-1$ is even and $m \mid a(b-1)$)

But wait – this gives us $m \mid (b-1)$ where $b$ satisfies $2b$
$\hookrightarrow$  $\equiv 3 \pmod{n}$.

Similarly, we get $n \mid (a-1)$ where $a$ satisfies $2a \equiv 3$
$\hookrightarrow$  $\pmod{m}$.

Let me check if any odd integer $m > 1$ can satisfy these
$\hookrightarrow$  conditions:

For $m = 3$: $a \equiv 3 \cdot 2^{-1} \equiv 3 \cdot 2 \equiv 0$
$\hookrightarrow$  $\pmod{3}$
So $a-1 \equiv -1 \equiv 2 \pmod{3}$
But we need $3 \mid (a-1)$, which means $a-1 \equiv 0 \pmod{3}$.
$\hookrightarrow$  Contradiction!

For $m = 5$: $a \equiv 3 \cdot 3 \equiv 4 \pmod{5}$ (since
$\hookrightarrow$  $2^{-1} \equiv 3 \pmod{5}$)
So $a-1 \equiv 3 \pmod{5}$
But we need $5 \mid (a-1)$, which means $a-1 \equiv 0 \pmod{5}$.
$\hookrightarrow$  Contradiction!

By similar analysis, no odd $m > 1$ can satisfy the required
$\hookrightarrow$  conditions.

```
84  Therefore $m = 1$, which means $y = 1$.
85
86  By symmetry, $n = 1$, which means $x = 1$.
87
88  Therefore $G = \{1\}$ is the trivial group.
89
```

## B.9 GROK 4 0709

Note: Grok 4 repeatedly returned empty output in OpenRouter. Thus Grok 4 0709 was accessed via LMArena. It is unclear if the two versions are identical.

Line 5: Nothing of the sort is established. (Critical)

Line 7: No reason is given for this. (Critical)

```
1  The group \(G\) is presented as \(\langle x, y \mid x y^2 = y^3
   ↪  x, \, y x^2 = x^3 y \rangle\).
2
3  Assume for contradiction that \(G\) is non-trivial. Then at least
   ↪  one of \(x\) or \(y\) is non-trivial.
4
5  As established, the orders \(n\) of \(y\) and \(m\) of \(x\) must
   ↪  be finite and odd.
6
7  The relations imply that conjugation by \(x\) induces an
   ↪  automorphism on \(\langle y \rangle\) multiplying exponents
   ↪  by \(k \equiv 3 \cdot 2^{-1} \pmod{n}\), with \(\gcd(k, n) =
   ↪  1\).
8
9  Similarly, conjugation by \(y\) induces an automorphism on
   ↪  \(\langle x \rangle\) multiplying exponents by \(l \equiv 3
   ↪  \cdot 2^{-1} \pmod{m}\), with \(\gcd(l, m) = 1\).
10
11 The order \(e\) of \(k\) modulo \(n\) divides \(m\) and \(l -
   ↪  1\).
12
13 The order \(d\) of \(l\) modulo \(m\) divides \(n\) and \(k -
   ↪  1\).
14
15 Now, \(k - 1 \equiv 3 \cdot 2^{-1} - 1 \pmod{n}\). Suppose an odd
   ↪  prime \(p\) divides both \(n\) and \(k - 1\). Then \(3 \cdot
   ↪  2^{-1} \equiv 1 \pmod{p}\), so \(3 \equiv 2 \pmod{p}\), hence
   ↪  \(p \mid 1\), impossible.
16
17 Thus, \(\gcd(n, k - 1) = 1\), so \(d = 1\).
18
19 Then \(l \equiv 1 \pmod{m}\), so \(2 \cdot 1 \equiv 3 \pmod{m}\),
   ↪  hence \(m \mid 1\), so \(m = 1\) and \(x = 1\).
20
21 With \(x = 1\), the first relation becomes \(y^2 = y^3\), so \(y
   ↪  = 1\), hence \(n = 1\).
22
23 This contradicts \(G\) non-trivial.
24
25 Therefore, \(G\) is trivial.
26
27 \boxed{G is trivial}
28
```

### B.10 KIMI K2

Note: The fact that at the end of the output generated by Kimi K2 a $\$\$$ is missing is not a typo. The model terminates its output without claiming to have found a solution. The error below occurs well before that time.

Line 184: The identity

$$y^3 = x^2 y^4 x^{-2}$$

does not follow by conjugating again by $x$, and in fact, does not follow from the previous line at all. (Critical)

```
1   ### Understanding the Problem
2
3   We have a group $$ G $$ defined by two generators $$ x $$ and $$
    ↪  y $$ with two relations:
4   1. $$ xy^2 = y^3x $$
5   2. $$ yx^2 = x^3y $$
6
7   Our goal is to prove that $$ G $$ is the trivial group, meaning
    ↪  $$ x = 1 $$ and $$ y = 1 $$ in $$ G $$, so $$ G $$ has only
    ↪  one element.
8
9   ### Initial Approach
10
11  To show that $$ G $$ is trivial, we need to derive from the given
    ↪  relations that $$ x = 1 $$ and $$ y = 1 $$. This means we
    ↪  should manipulate the given equations to find simpler
    ↪  relationships or equalities that force $$ x $$ and $$ y $$ to
    ↪  be the identity.
12
13  ### Manipulating the First Relation: $$ xy^2 = y^3x $$
14
15  Let's start with the first relation:
16  $$ xy^2 = y^3x $$
17
18  We can multiply both sides on the right by $$ x^{-1} $$ (the
    ↪  inverse of $$ x $$) to isolate terms:
19  $$ xy^2x^{-1} = y^3 $$
20
21  This shows that $$ y^3 $$ is equal to $$ x y^2 x^{-1} $$, which
    ↪  is the conjugation of $$ y^2 $$ by $$ x $$.
22
23  ### Manipulating the Second Relation: $$ yx^2 = x^3y $$
24
25  Similarly, take the second relation:
26  $$ yx^2 = x^3y $$
27
28  Multiply both sides on the right by $$ y^{-1} $$:
29  $$ yx^2y^{-1} = x^3 $$
30
31  Here, $$ x^3 $$ is the conjugation of $$ x^2 $$ by $$ y $$.
32
33  ### Exploring Conjugation and Powers
34
35  From the first manipulation, we have:
36  $$ x y^2 x^{-1} = y^3 $$
37
```

This implies that $$ x $$ conjugates $$ y^2 $$ to $$ y^3 $$.
Similarly, from the second manipulation:
$$ y x^2 y^{-1} = x^3 $$

So $$ y $$ conjugates $$ x^2 $$ to $$ x^3 $$.

### Trying to Find Simpler Relations

Let's see if we can find a relationship between $$ x $$ and $$ y $$ that leads to $$ x = 1 $$ or $$ y = 1 $$.

From $$ x y^2 x^{-1} = y^3 $$, we can raise both sides to the power of $$ n $$ (for any integer $$ n $$):
$$ (x y^2 x^{-1})^n = y^{3n} $$
$$ x y^{2n} x^{-1} = y^{3n} $$

Similarly, from $$ y x^2 y^{-1} = x^3 $$:
$$ (y x^2 y^{-1})^n = x^{3n} $$
$$ y x^{2n} y^{-1} = x^{3n} $$

These show how conjugation by $$ x $$ and $$ y $$ affects powers of $$ y $$ and $$ x $$, respectively.

### Attempting to Find a Fixed Point

Let's consider the exponent of $$ y $$ in the first relation. The exponents on both sides of $$ x y^2 x^{-1} = y^3 $$ don't match directly, which is unusual. This suggests that the only way this can hold is if $$ y = 1 $$, making both sides equal to 1.

Similarly, if $$ y = 1 $$, then plugging into the second relation:
$$ 1 \cdot x^2 \cdot 1 = x^3 \cdot 1 $$
$$ x^2 = x^3 $$
$$ x = 1 $$

By symmetry, if we assume $$ x = 1 $$, then:
First relation:
$$ 1 \cdot y^2 = y^3 \cdot 1 $$
$$ y^2 = y^3 $$
$$ y = 1 $$

This suggests that $$ x = y = 1 $$ is a solution that satisfies both relations, making $$ G $$ trivial. But we need to ensure that this is the only possible solution within the group $$ G $$.

### Proving Uniqueness of Trivial Solution

Let's formalize this without assuming $$ x $$ or $$ y $$ is the identity upfront.

From $$ x y^2 x^{-1} = y^3 $$, we can consider the exponent as a kind of homomorphism. Specifically, the mapping that sends $$ y $$ to $$ y^3 $$ when conjugated by $$ x $$. Similarly, the other relation gives a mapping sending $$ x $$ to $$ x^3 $$ under conjugation by $$ y $$.

This is reminiscent of the concept where in a group, if conjugation by an element raises powers to higher exponents, the group might be forced to be trivial.

Let's try to express everything in terms of a single element. Suppose we express one element in terms of the other.

From $$ x y^2 x^{-1} = y^3 $$, we can write:
$$ x y^2 = y^3 x $$

Similarly, from $$ y x^2 y^{-1} = x^3 $$:
$$ y x^2 = x^3 y $$

Now, let's try to find a word that must equal the identity.

Compute $$ x y x^{-1} y^{-1} $$, but this might not directly help. Instead, let's try to find a power that collapses to identity.

### Using Commutators

Notice that the relations resemble "expansive" behaviour-conjugation increases exponents. In some contexts, this implies the group must be trivial unless the elements commute in a certain way.

Let's attempt to compute some commutators or express higher powers.

From $$ x y^2 x^{-1} = y^3 $$, we can apply this multiple times:

First, raise both sides to the power of $$ k $$:
$$ (x y^2 x^{-1})^k = y^{3k} $$
$$ x y^{2k} x^{-1} = y^{3k} $$

Now, set $$ k = 0 $$, but that gives $$ 1 = 1 $$, which is trivial.

Alternatively, express $$ y $$ in terms of itself:
$$ x y^2 x^{-1} = y^3 $$
$$ x y x^{-1} x y x^{-1} = y^3 $$
$$ (x y x^{-1})^2 = y^3 $$

Similarly, from $$ y x^2 y^{-1} = x^3 $$:
$$ (y x y^{-1})^2 = x^3 $$

This seems to imply that the conjugates have certain power relationships.

### Trying Specific Elements and Equalities

Let's consider the order of $$ x $$ and $$ y $$. Suppose $$ x $$ has finite order $$ n $$, then $$ x^n = 1 $$.

From $$ y x^2 y^{-1} = x^3 $$, we have:
$$ y x^2 y^{-1} = x^3 $$
$$ y x^4 y^{-1} = x^6 $$
$$ y x^{2k} y^{-1} = x^{3k} $$

For $$ k = n $$, since $$ x^n = 1 $$, we have:
$$ y 1 y^{-1} = 1 = x^{3n} $$
But $$ x^{3n} = (x^n)^3 = 1^3 = 1 $$, which doesn't give new information.

Similarly, from $$ x y^2 x^{-1} = y^3 $$, we have:
$$ x y^{2k} x^{-1} = y^{3k} $$

If $$ y $$ has finite order $$ m $$, then for $$ k = m $$:
$$ x 1 x^{-1} = 1 = y^{3m} $$
$$ y^{3m} = 1 $$

But since the order of $$ y $$ is $$ m $$, $$ m $$ divides $$ 3m $$, which is always true, so no contradiction arises directly from this.

### Introducing a Normal Subgroup Approach

Perhaps we can consider that these relations imply all elements are trivial. Let's try to prove that $$ x $$ and $$ y $$ commute, and then see if this forces them to be trivial.

Assume $$ xy = yx $$. Then from $$ xy^2 = y^3x $$:
$$ xy^2 = y^3x $$
$$ y^2x = y^3x $$ (since $$ x $$ and $$ y $$ commute)
Multiply both sides on the right by $$ x^{-1} y^{-2} $$:
$$ 1 = y $$

Similarly, from $$ yx^2 = x^3y $$:
$$ yx^2 = x^3y $$
$$ x^2y = x^3y $$
$$ x^2 = x^3 $$
$$ x = 1 $$

Thus, if $$ x $$ and $$ y $$ commute, both must be identity. But we don't know if they commute.

### Using the Composition of Conjugations

We have:
$$ \phi_x: y \mapsto x y x^{-1} $$
with $$ \phi_x(y^2) = y^3 $$, so $$ (\phi_x(y))^2 = y^3 $$, implying $$ \phi_x(y) $$ is a square root of $$ y^3 $$, i.e., $$ x y x^{-1} = y^{3/2} $$, but this suggests fractional exponents, which are typically not in groups.

This seems problematic unless $$ y = 1 $$.

Let me rephrase: from $$ x y^2 x^{-1} = y^3 $$, we can define $$ z = x y x^{-1} $$, then:
$$ z^2 = y^3 $$

Similarly, from $$ y x^2 y^{-1} = x^3 $$, let $$ w = y x y^{-1} $$, then:
$$ w^2 = x^3 $$

Now we have:
$$ x y x^{-1} = z $$ where $$ z^2 = y^3 $$
$$ y x y^{-1} = w $$ where $$ w^2 = x^3 $$

This seem to imply that $$ z $$ is some root of $$ y^3 $$ and $$ w $$ of $$ x^3 $$, but in groups, this often forces elements to be identity.

### Deriving Contradictions or Identities

Let's try to express everything in terms of $$ x $$ and its conjugates.

From $$ x y^2 x^{-1} = y^3 $$, we have:
$$ y^3 = x y^2 x^{-1} $$
$$ y^3 = x^2 y^4 x^{-2} $$ (conjugating by $$ x $$ again)
$$ y^3 = x^2 y^4 x^{-2} $$

But also:
$$ y^3 = x y^2 x^{-1} $$
So:
$$ x y^2 x^{-1} = x^2 y^4 x^{-2} $$
$$ y^2 = x y^4 x^{-1} $$
But $$ x y^4 x^{-1} = (x y^2 x^{-1})^2 = (y^3)^2 = y^6 $$, so:
$$ y^2 = y^6 $$
$$ y^4 = 1 $$

Similarly, from $$ y x^2 y^{-1} = x^3 $$, applying $$ y $$ twice:
$$ y^2 x^4 y^{-2} = x^9 $$
But also:
$$ x^3 = y x^2 y^{-1} $$
So:
$$ y x^2 y^{-1} = y^2 x^4 y^{-2} $$
$$ x^2 = y x^4 y^{-1} $$
$$ x^2 = (x^3)^2 = x^6 $$
$$ x^4 = 1 $$

Now we have $$ x^4 = 1 $$

### B.11 QWEN3 235B A22B THINKING 2507

`Line 25`: The model has assumed commutativity. (Critical)

```
1  To determine whether the group $ G = \langle x, y \mid xy^2 =
   ↪  y^3x,\ yx^2 = x^3y \rangle $ is trivial, we analyze the given
   ↪  relations and manipulate them using group-theoretic
   ↪  techniques.
2
3  ---
4
5  ### Step 1: Expressing Generators in Terms of a Single Element
6
7  Define the **commutator** $ a = [x, y] = x y x^{-1} y^{-1} $.
   ↪  Then by definition:
8
9  $$
10 x y = a y x.
11 $$
12
13 Now apply this to **Relation 1**:
14 $$
15 xy^2 = y^3x.
16 $$
17
18 Left-hand side:
19 $$
20 x y^2 = x y \cdot y = (a y x) \cdot y = a y x y.
21 $$
22
23 Using $ x y = a y x $ again inside:
24 $$
25 a y x y = a y (a y x) = a^2 y^2 x.
26 $$
27
28 So:
29 $$
30 x y^2 = a^2 y^2 x.
31 $$
32
33 Equating with the right-hand side $ y^3 x $, we get:
34 $$
35 a^2 y^2 x = y^3 x \Rightarrow a^2 y^2 = y^3 \Rightarrow a^2 = y.
36 $$
37
38 Thus, we have:
39 $$
40 y = a^2. \tag{1}
41 $$
42
43 ---
44
45 ### Step 2: Using the Second Relation
46
47 Now consider **Relation 2**:
48 $$
```

```
yx^2 = x^3 y.
$$

We define the inverse commutator $ b = [y, x] = y x y^{-1} x^{-1}
  = a^{-1} $, so:
$$
y x = b x y = a^{-1} x y.
$$

Apply this to Relation 2:
$$
y x^2 = y x \cdot x = (a^{-1} x y) \cdot x = a^{-1} x y x.
$$

Then:
$$
y x^2 = a^{-1} x (a^{-1} x y) = a^{-2} x^2 y.
$$

Equating with RHS $ x^3 y $, we get:
$$
a^{-2} x^2 y = x^3 y \Rightarrow a^{-2} x^2 = x^3 \Rightarrow x =
  a^{-2}. \tag{2}
$$

---

### Step 3: Substituting Back into Relations

From (1) and (2), we now express both generators in terms of $ a
  $:
$$
x = a^{-2}, \quad y = a^2.
$$

Substitute these into **Relation 1**:
$$
xy^2 = a^{-2}(a^2)^2 = a^{-2} a^4 = a^2,
$$
$$
y^3 x = (a^2)^3 a^{-2} = a^6 a^{-2} = a^4.
$$

So:
$$
a^2 = a^4 \Rightarrow a^2 = e.
$$

Therefore, $ a = e $, which implies:
$$
x = a^{-2} = e, \quad y = a^2 = e.
$$

---

### Step 4: Verifying Triviality
```

```
102
103  With $ x = e $ and $ y = e $, all relations are trivially
     ↪   satisfied:
104  - $ xy^2 = e \cdot e = e $, and $ y^3x = e \cdot e = e $,
105  - $ yx^2 = e \cdot e = e $, and $ x^3y = e \cdot e = e $.
106
107  Hence, the only solution is the trivial group.
108
109  ---
110
111  ### Conclusion
112
113  By expressing the generators in terms of a single element derived
     ↪   from the commutator and analyzing the resulting constraints,
     ↪   we find that the relations force both $ x $ and $ y $ to be
     ↪   the identity element. Therefore, the group defined by the
     ↪   presentation:
114
115  $$
116  G = \langle x, y \mid xy^2 = y^3x,\ yx^2 = x^3y \rangle
117  $$
118
119  is **trivial**.
120
121  $$
122  \boxed{G \text{ is the trivial group}}
123  $$
124
```

## B.12   GLM-4.5

Note: The generation began in the middle of the sentence, our reproduction below is not a typo.

Line 19: This identity is incorrect. The model has made an algebra error from the previous line. (Critical)

```
1   that the group $G = \langle x, y \mid xy^2 = y^3x, yx^2 = x^3y
    ↪   \rangle$ is the trivial group, I'll derive that both
    ↪   generators $x$ and $y$ must be the identity element.
2
3   Starting with the given relations:
4   1. $xy^2 = y^3x$
5   2. $yx^2 = x^3y$
6
7   From relation (1), multiply both sides by $y^{-2}$ on the left:
8   $y^{-2}xy^2 = yx$   (3)
9
10  From relation (2), multiply both sides by $x^{-2}$ on the left:
11  $x^{-2}yx^2 = xy$   (4)
12
13  Let $A = y^{-1}xy$ and $B = x^{-1}yx$. Then:
14  - From (3): $y^{-1}Ay = yx$
15  - From (4): $x^{-1}Bx = xy$
16
17  Since $A = y^{-1}xy$, we have $y^{-1}Ay = y^{-2}xy^2$. From
    ↪   equation (3), $y^{-2}xy^2 = yx$, so:
```

```
18  $y^{-1}Ay = yx$
19  which implies $A = y^2x$.
20
21  Similarly, since $B = x^{-1}yx$, we have $x^{-1}Bx = x^{-2}yx^2$.
    ↪  From equation (4), $x^{-2}yx^2 = xy$, so:
22  $x^{-1}Bx = xy$
23  which implies $B = x^2y$.
24
25  Now we have:
26  $A = y^{-1}xy = y^2x$
27  $B = x^{-1}yx = x^2y$
28
29  From $yA = xy$ and $A = y^2x$, we get:
30  $y(y^2x) = xy$
31  $y^3x = xy$
32
33  From the original relation (1), $xy^2 = y^3x$, so:
34  $xy^2 = xy$
35
36  Now, multiply both sides of $xy^2 = xy$ by $x^{-1}$ on the left:
37  $y^2 = y$
38
39  Multiply both sides of $y^2 = y$ by $y^{-1}$ on the right:
40  $y = e$
41
42  Since $y$ is the identity element, we can substitute $y = e$ into
    ↪  relation (1):
43  $xe^2 = e^3x$
44  $x = x$
45
46  This is a tautology and doesn't give us new information. Let's
    ↪  substitute $y = e$ into relation (2):
47  $ex^2 = x^3e$
48  $x^2 = x^3$
49
50  Multiply both sides of $x^2 = x^3$ by $x^{-2}$ on the right:
51  $e = x$
52
53  Therefore, both $x$ and $y$ are the identity element, which means
    ↪  $G$ is the trivial group.
```

## B.13  GEMINI 2.5 DEEP THINK

Line 18: The model claims $yx^3 = (yx)x^3$, which is an algebra error. (Critical)

```
1   Yes, we can prove that G is the trivial group.
2
3   Let $G$ be a group with generators $x$ and $y$ and the relations:
4   1) $xy^2 = y^3x$
5   2) $yx^2 = x^3y$
6
7   We want to show that $x=e$ and $y=e$, where $e$ is the identity
    ↪   element.
8
9   Let's start by manipulating the second relation (2). We can
    ↪   rewrite it as:
10  $x^3y = yx^2$ (2')
11
12  Now, we examine how the element $x^3$ interacts with the element
    ↪   $yx$. We look at the product $x^3(yx)$:
13  $x^3(yx) = (x^3y)x$ (by associativity)
14
15  We use the rearranged relation (2') to substitute $x^3y$ with
    ↪   $yx^2$:
16  $(x^3y)x = (yx^2)x = y(x^2x) = yx^3$
17
18  So we have found that $x^3(yx) = yx^3$. Since $yx^3 = (yx)x^3$,
    ↪   this means that $x^3$ commutes with $yx$. In terms of
    ↪   commutators, we have $[x^3, yx] = e$.
19
20  Next, let's analyze the commutator $[x^3, yx]$. Recall that the
    ↪   commutator $[A, B]$ is defined as $ABA^{-1}B^{-1}$.
21  $[x^3, yx] = x^3 (yx) (x^3)^{-1} (yx)^{-1}$
22  $= x^3 y x x^{-3} (x^{-1}y^{-1})$
23  $= x^3 y (x^{1-3-1}) y^{-1}$
24  $= x^3 y x^{-3} y^{-1}$
25
26  This expression is exactly the definition of the commutator of
    ↪   $x^3$ and $y$, i.e., $[x^3, y]$.
27
28  Since we established that $[x^3, yx] = e$ and $[x^3, yx] = [x^3,
    ↪   y]$, we must have:
29  $[x^3, y] = e$.
30
31  This means that $x^3$ commutes with $y$:
32  $x^3y = yx^3$.
33
34  Now we have two equations involving $x^3y$:
35  From relation (2'): $x^3y = yx^2$
36  From our derivation: $x^3y = yx^3$
37
38  We can equate the right-hand sides:
39  $yx^2 = yx^3$
40
41  We multiply by $y^{-1}$ on the left:
42  $x^2 = x^3$
43
```

```
44  We multiply by $x^{-2}$ (on either side):
45  $e = x$.
46
47  Now that we have shown $x=e$, we substitute this back into the
    ↪  first relation (1):
48  $xy^2 = y^3x$
49  $e y^2 = y^3 e$
50  $y^2 = y^3$
51
52  We multiply by $y^{-2}$:
53  $e = y$.
54
55  Since both generators $x$ and $y$ are equal to the identity
    ↪  element, the group $G$ is the trivial group.
56
57
```

### B.14 LLAMA 4 MAVERICK

Lines 4-8: A lot of effort is expended proving $y^3x = y^3x$.

Line 11: The model claims $x(y^2)x^{-1}(y^2)^{-1} = e$ but never proves this. (Critical)

```
## Step 1: Understand the given problem and the relations
↪  provided for the group $G$.
The group $G$ is generated by $x$ and $y$ with the relations
↪  $xy^2 = y^3x$ and $yx^2 = x^3y$. We need to determine if $G$
↪  is the trivial group, meaning it contains only the identity
↪  element.

## Step 2: Manipulate the given relations to simplify or derive
↪  useful equations.
Starting with $xy^2 = y^3x$, we can rearrange this to $xy^2x^{-1}
↪  = y^3$, implying that $x$ conjugates $y^2$ to $y^3$.
↪  Similarly, from $yx^2 = x^3y$, we get $yxy^{-1} = x^3x^{-2} =
↪  x$, showing that $y$ conjugates $x$ to $x$.

## Step 3: Further simplify the relations to check for any
↪  patterns or simplifications that could lead to proving $G$ is
↪  trivial.
From $xy^2 = y^3x$, we can also derive $x^{-1}y^3x = y^2$. This
↪  implies $x^{-1}y^3x = y^2$ and thus $y^3 = xy^2x^{-1}$.
↪  Substituting $y^2 = x^{-1}y^3x$ into the original equation
↪  yields $x(x^{-1}y^3x) = y^3x$, simplifying to $y^3x = y^3x$,
↪  which is always true and doesn't provide new information
↪  directly.

## Step 4: Analyze the relation $xy^2 = y^3x$ further to derive a
↪  condition on $y$.
Rearranging $xy^2 = y^3x$ gives $xy^2x^{-1}y^{-3} = e$, where $e$
↪  is the identity. This can be written as
↪  $xy^2x^{-1}y^{-2}y^{-1} = e$, or
↪  $x(y^2)x^{-1}(y^2)^{-1}y^{-1} = e$, simplifying to $y^{-1} =
↪  e$ because $x(y^2)x^{-1}(y^2)^{-1} = e$ implies $y^2 = e$
↪  after simplification, but directly we see $y = e$ from
↪  $y^{-1} = e$.

## Step 5: Examine the implication of $y = e$ on the group $G$
↪  and its generators.
If $y = e$, then substituting into one of the original relations,
↪  $xy^2 = y^3x$ becomes $x = x$, which is true for any $x$.
↪  However, we need to check the other relation as well. $yx^2 =
↪  x^3y$ becomes $ex^2 = x^3e$, simplifying to $x^2 = x^3$. This
↪  implies $x^3x^{-2} = e$, or $x = e$.

## Step 6: Verify the result by checking if both generators being
↪  the identity implies $G$ is trivial.
Since both $x = e$ and $y = e$, and $G$ is generated by $x$ and
↪  $y$, $G$ must be the trivial group because it is generated by
↪  the identity.

The final answer is: $\boxed{Yes}$
```

### B.15    DEEPSEEK V3 0324

`Line 46`: The order is only a positive integer if it is finite. The model has ignored the infinite case. However, it recovers from this on line 62.

`Line 82`: The model worries that $x^3 = x^2$ can only be simplified to $x = e$ if $x$ is cancellable. But this is an unfounded concern, as we are working in a group.

`Line 194`: The model makes no progress with the problem, other than showing $|x|$ and $|y|$ are coprime to 6, but then concludes the group is trivial. No part of the proof comes close to showing this. (Critical)

```
1  Alright, let's tackle this problem step by step. We have a group
   ↪   $$ G $$ generated by two elements $$ x $$ and $$ y $$ with
   ↪   the following relations:
2
3  1. $$ xy^2 = y^3x $$
4  2. $$ yx^2 = x^3y $$
5
6  Our goal is to show that $$ G $$ is the trivial group, meaning
   ↪   that $$ x $$ and $$ y $$ are both the identity element, and
   ↪   thus $$ G = \{e\} $$.
7
8  ### Understanding the Problem
9
10 First, let's recall what these relations mean. In a group
   ↪   presentation, relations are equations that hold true among
   ↪   the generators. Here, the two relations give us ways to
   ↪   manipulate expressions involving $$ x $$ and $$ y $$.
11
12 Our strategy will be to derive consequences from these relations,
   ↪   hoping to find that $$ x $$ and $$ y $$ must be the identity.
13
14 ### Starting with the First Relation: $$ xy^2 = y^3x $$
15
16 Let's see what this tells us. If we multiply both sides on the
   ↪   left by $$ x^{-1} $$, we get:
17
18 $$ x^{-1}xy^2 = x^{-1}y^3x $$
19
20 Which simplifies to:
21
22 $$ y^2 = x^{-1}y^3x $$
23
24 This can be rewritten as:
25
26 $$ x y^2 x^{-1} = y^3 \tag{Equation 1}$$
27
28 This suggests that conjugating $$ y^2 $$ by $$ x $$ gives $$ y^3
   ↪   $$.
29
30 Similarly, from the second relation $$ yx^2 = x^3y $$, let's
   ↪   multiply both sides on the left by $$ y^{-1} $$:
31
32 $$ y^{-1}yx^2 = y^{-1}x^3y $$
33
34 Simplifying:
35
```

$$ x^2 = y^{-1}x^3y $$

Which can be rewritten as:

$$ y x^2 y^{-1} = x^3 \tag{Equation 2}$$

This suggests that conjugating $$ x^2 $$ by $$ y $$ gives $$ x^3 $$.

### Exploring Equation 1: $$ x y^2 x^{-1} = y^3 $$

This equation tells us that $$ y^2 $$ and $$ y^3 $$ are conjugates, implying that they have the same order in the group. The order of an element $$ g $$, denoted $$ |g| $$, is the smallest positive integer $$ n $$ such that $$ g^n = e $$.

From $$ y^2 $$ and $$ y^3 $$ being conjugates:

$$ |y^2| = |y^3| $$

We know that in general, $$ |g^k| = \frac{|g|}{\gcd(|g|, k)} $$. So:

$$ \frac{|y|}{\gcd(|y|, 2)} = \frac{|y|}{\gcd(|y|, 3)} $$

This implies:

$$ \gcd(|y|, 2) = \gcd(|y|, 3) $$

The only way this can happen is if $$ \gcd(|y|, 6) = 1 $$, meaning that $$ |y| $$ is coprime with 6. The simplest possibility is $$ |y| = 1 $$, i.e., $$ y = e $$.

But let's not jump to conclusions; maybe $$ |y| $$ is infinite or another number coprime to 6. Let's explore further.

### Exploring Equation 2: $$ y x^2 y^{-1} = x^3 $$

Similarly, this tells us that $$ x^2 $$ and $$ x^3 $$ are conjugates, so:

$$ |x^2| = |x^3| $$

Again, using the order formula:

$$ \frac{|x|}{\gcd(|x|, 2)} = \frac{|x|}{\gcd(|x|, 3)} $$

Thus:

$$ \gcd(|x|, 2) = \gcd(|x|, 3) $$

Similarly, this implies $$ \gcd(|x|, 6) = 1 $$, so $$ |x| $$ is coprime with 6. Again, the simplest solution is $$ |x| = 1 $$, i.e., $$ x = e $$.

### Considering $$ x $$ and $$ y $$ are the Identity

If $$ y = e $$, then the first relation $$ xy^2 = y^3x $$ becomes $$ x = x $$, which holds trivially. The second relation $$ yx^2 = x^3y $$ becomes $$ x^2 = x^3 $$, which implies $$ x = e $$ (assuming $$ x $$ has finite order or is cancellable).

Similarly, if $$ x = e $$, the second relation becomes $$ y = y $$, and the first relation becomes $$ y^2 = y^3 $$, implying $$ y = e $$.

This suggests that the only solution is $$ x = y = e $$.

### Trying to Find Non-Trivial Solutions

But let's not assume the group is finite yet. Maybe the group is infinite, and $$ x $$ and $$ y $$ have infinite order. Let's see if we can find any non-trivial relations.

From $$ x y^2 x^{-1} = y^3 $$, we can think of this as an action of $$ x $$ on $$ y $$. In general, this could suggest a semidirect product, but the relations seem too restrictive.

Similarly, $$ y x^2 y^{-1} = x^3 $$ is an action of $$ y $$ on $$ x $$.

Perhaps we can find a consistency condition by combining these.

Let's see if we can express both $$ x $$ and $$ y $$ in terms of one another.

From $$ x y^2 = y^3 x $$, we can try to solve for $$ y $$ in terms of $$ x $$, but it's not straightforward.

Alternatively, let's see if we can find that $$ x $$ and $$ y $$ commute.

Assume for a moment that $$ x $$ and $$ y $$ commute, i.e., $$ xy = yx $$. Then:

From $$ xy^2 = y^3x $$, with commutativity:

$$ y^2x = y^3x $$

Cancel $$ x $$ on the right (assuming $$ x $$ is cancellable, which it is in a group):

$$ y^2 = y^3 $$

Thus $$ y = e $$.

Similarly, from $$ yx^2 = x^3y $$, with commutativity:

$$ x^2y = x^3y $$

Cancel $$ y $$:

$$ x^2 = x^3 $$

Thus $$ x = e $$.

So if $$ x $$ and $$ y $$ commute, they must both be the identity.

But we haven't shown that $$ x $$ and $$ y $$ must commute. Maybe the relations force them to commute.

Let's attempt to see if $$ xy = yx $$ follows from the given relations.

From $$ xy^2 = y^3x $$, let's try to express $$ xy $$:

Multiply both sides by $$ y^{-1} $$ on the right:

$$ xy = y^3x y^{-1} $$

Similarly, from $$ yx^2 = x^3y $$, multiply by $$ x^{-1} $$ on the right:

$$ yx = x^3y x^{-1} $$

This seems circular. Maybe another approach is better.

### Attempting to Find Orders

Let me think about the orders of $$ x $$ and $$ y $$. From earlier, $$ |y^2| = |y^3| $$, which implies $$ \gcd(|y|, 2) = \gcd(|y|, 3) $$.

Possible orders for $$ y $$:

If $$ |y| $$ is finite, then $$ \gcd(|y|, 2) = \gcd(|y|, 3) $$. The numbers where this holds are those not divisible by 2 or 3 (or both), i.e., numbers coprime to 6: 1, 5, 7, 11, etc.

Similarly, $$ |x| $$ must be coprime to 6.

The simplest case is $$ |x| = |y| = 1 $$, i.e., trivial group.

If $$ |y| > 1 $$, say $$ |y| = 5 $$, then $$ y^2 $$ has order $$ 5 / \gcd(5,2) = 5 $$, and $$ y^3 $$ has order $$ 5 / \gcd(5,3) = 5 $$, so the equation holds. But does this lead to a contradiction elsewhere?

From $$ x y^2 x^{-1} = y^3 $$, if $$ y $$ has order 5, then $$ y^3 = y^{-2} $$, so:

$$ x y^2 x^{-1} = y^{-2} $$

But $$ y^2 $$ has order 5, and its inverse also has order 5, so this is possible, but doesn't immediately force $$ y = e $$.

```
164  Similarly, for $$ x $$, if $$ |x| = 5 $$, from $$ y x^2 y^{-1} =
     ↪   x^3 $$, and $$ x^3 = x^{-2} $$, so:
165
166  $$ y x^2 y^{-1} = x^{-2} $$
167
168  Again, no immediate contradiction.
169
170  However, the group seems too constrained to allow non-trivial
     ↪   elements, but I'm not seeing an immediate contradiction for
     ↪   higher orders coprimes to 6.
171
172  ### Trying Specific Cases
173
174  Perhaps assume $$ y = e $$, then see what $$ x $$ must be.
175
176  If $$ y = e $$, then relations become:
177
178  1. $$ x = x $$ (holds)
179  2. $$ x^2 = x^3 \implies x = e $$
180
181  So $$ x = e $$, $$ y = e $$. Trivial group.
182
183  Similarly, if $$ x = e $$, then relations become:
184
185  1. $$ y^2 = y^3 \implies y = e $$
186  2. $$ y = y $$ (holds)
187
188  Again, trivial group.
189
190  This suggests that the only solution is the trivial group.
191
192  ### Conclusion
193
194  After exploring these relations, it appears that the only group
     ↪   $$ G $$ satisfying both $$ xy^2 = y^3x $$ and $$ yx^2 = x^3y
     ↪   $$ is the trivial group where both generators $$ x $$ and $$
     ↪   y $$ are the identity element. Any non-trivial assignment
     ↪   leads to contradictions or forces the generators to be the
     ↪   identity.
195
196  ### Final Answer
197
198  After carefully analyzing the given relations, we conclude that
     ↪   the group $$ G $$ must be the trivial group. That is, $$ G =
     ↪   \{e\} $$, where $$ e $$ is the identity element, and $$ x = y
     ↪   = e $$.
199
200  ### Verification
201
202  Let's verify:
203
204  1. $$ xy^2 = e \cdot e^2 = e $$ and $$ y^3x = e^3 \cdot e = e $$,
     ↪   so $$ xy^2 = y^3x $$.
205  2. $$ yx^2 = e \cdot e^2 = e $$ and $$ x^3y = e^3 \cdot e = e $$,
     ↪   so $$ yx^2 = x^3y $$.
206
```

```
207  Both relations hold, and no smaller group satisfies them unless
     ↪  both generators are the identity. Hence, $$ G $$ is indeed
     ↪  the trivial group.
208
```

### B.16   QwQ 32B

`Lines 7-9`: There is an algebra error in deriving $x^{-1}y^3xy^{-3} = e$. (Critical)

`Line 13`: Commutativity is incorrectly assumed. (Critical)

`Line 25`: Commutativity is incorrectly assumed. (Critical)

```
1  To prove that the group \( G \) generated by \( x \) and \( y \)
   ↪  with relations \( xy^2 = y^3x \) and \( yx^2 = x^3y \) is
   ↪  trivial, we start by analyzing the given relations.
2
3  First, consider the relation \( xy^2 = y^3x \). Rearranging this,
   ↪  we have:
4  \[
5  x^{-1} y^3 x = y^2
6  \]
7  Multiplying both sides on the right by \( y^{-3} \), we get:
8  \[
9  x^{-1} y^3 x y^{-3} = e
10 \]
11 This simplifies to:
12 \[
13 x^{-1} (y^3 y^{-3}) x = e \implies x^{-1} y^0 x = e \implies
   ↪  x^{-1} e x = e
14 \]
15 However, we need to re-express this correctly. Starting again:
16 \[
17 x y^2 x^{-1} = y^3
18 \]
19 Multiply both sides by \( y^{-3} \):
20 \[
21 x y^2 x^{-1} y^{-3} = e
22 \]
23 This can be rewritten as:
24 \[
25 x (y^{-1}) x^{-1} = e
26 \]
27 Multiplying both sides on the left by \( x^{-1} \) and on the
   ↪  right by \( x \), we get:
28 \[
29 y^{-1} = e \implies y = e
30 \]
31 Thus, \( y \) must be the identity element.
32
33 Substituting \( y = e \) into the second relation \( yx^2 = x^3y
   ↪  \), we get:
34 \[
35 e \cdot x^2 = x^3 \cdot e \implies x^2 = x^3
36 \]
37 This simplifies to:
38 \[
39 x^2 = x^3 \implies x^2 x^{-2} = x^3 x^{-2} \implies e = x
40 \]
41 Therefore, \( x \) must also be the identity element.
42
```

```
43  Since both \( x \) and \( y \) are the identity element, the
    ↪   group \( G \) is trivial.
44
45  \[
46  \boxed{1}
47  \]
48
```

### B.17   GPT-OSS-120B

`Line 24`: The given conjugation relations only show that the subgroups generated by $y^2$ and $x^2$ are normal, not those generated by $x$ and $y$. (Critical)

```
1  $$
2  G=\langle x ,y\mid xy^{2}=y^{3}x ,\; yx^{2}=x^{3}y\rangle .
3  $$
4
5  ________________________________________________________________⌋
   ↪  ----
6  ### 1.  Conjugation relations
7
8  From the defining equalities we can rewrite
9
10 $$
11 \begin{aligned}
12 xy^{2}&=y^{3}x \quad\Longrightarrow\quad
   ↪  x\,y^{2}x^{-1}=y^{3},\\$$2mm]
13 yx^{2}&=x^{3}y \quad\Longrightarrow\quad y\,x^{2}y^{-1}=x^{3}.
14 \end{aligned}
15 \tag{1}
16 $$
17
18 Thus
19
20 *conjugation by $$x$$ sends $$y^{2}$$ to $$y^{3}$$;*
21
22 *conjugation by $$y$$ sends $$x^{2}$$ to $$x^{3}$$.*
23
24 Hence the subgroups $$\langle y\rangle$$ and $$\langle x\rangle$$
   ↪  are normal, and the
25 conjugation maps restrict to automorphisms of these cyclic
   ↪  subgroups.
26
27 ________________________________________________________________⌋
   ↪  ----
28 ### 2.  The automorphisms are given by a power map
29
30 Let
31
32 $$
33 x y x^{-1}=y^{\ell},\qquad y x y^{-1}=x^{k}
34 \tag{2}
35 $$
36
37 for some integers $$\ell ,k$$.
38 Because the restriction of conjugation to a cyclic group is an
   ↪  automorphism,
39 $$\gcd(\ell ,|y|)=\gcd(k ,|x|)=1$$ (if the orders are infinite
   ↪  the condition is
40 $$\ell ,k\neq0$$).
41
42 Apply the first equality of (2) to the element $$y^{2}$$:
43 $$
44 x y^{2}x^{-1} = (x y x^{-1})^{2}=y^{2\ell}=y^{3}.
```

```
45  $$
46  Hence
47
48  $$
49  2\ell\equiv 3\pmod{|y|}\qquad\text{(or }2\ell=3\text{ if
    ↪   }|y|=\infty).
50  \tag{3}
51  $$
52
53  Similarly, applying the second equality of (2) to $$x^{2}$$ gives
54
55  $$
56  y x^{2}y^{-1}= (y x y^{-1})^{2}=x^{2k}=x^{3},
57  $$
58  so
59
60  $$
61  2k\equiv 3\pmod{|x|}\qquad\text{(or }2k=3\text{ if }|x|=\infty).
62  \tag{4}
63  $$
64
65  Because the congruences (3) and (4) have a solution, the orders
    ↪   of $$x$$ and
66  $$y$$ cannot be even and cannot be multiples of $$3$$; in
    ↪   particular they are
67  odd (or infinite).  In any case the integers $$\ell ,k$$ are well
    ↪   defined and
68  satisfy $$\ell\not\equiv 0$$ and $$k\not\equiv 0$$.
69
70  ----------------------------------------------------------------┘
    ↪   ----
71  ### 3.  $$x$$ and $$y$$ lie in the same cyclic subgroup
72
73  From (2) we compute the commutator:
74
75  $$
76  \begin{aligned}
77  x y x^{-1}y^{-1}
78     &= (x y x^{-1})y^{-1}
79        = y^{\ell}y^{-1}
80        = y^{\ell-1},\$$2mm]
81     &= x (y x y^{-1}) x^{-1}
82        = x x^{k}x^{-1}
83        = x^{k}.
84  \end{aligned}
85  $$
86
87  Thus
88
89  $$
90  x^{k}=y^{\ell-1}. \tag{5}
91  $$
92
93  Since $$k\not\equiv0$$ (mod $$|x|$$) the element $$x^{k}$$ is a
    ↪   non-trivial power
```

of $$x$$; (5) shows that this power is a power of $$y$$.
↪   Consequently

$$
x\in\langle y\rangle .
$$

A symmetric computation (or simply interchanging the roles of
↪   $$x$$ and $$y$$)
gives

$$
y\in\langle x\rangle .
$$

Hence both generators belong to the same cyclic subgroup; the
↪   whole group
$$G$$ is cyclic.

----------------------------------------------------------------┘
↪   ----
### 4.  A cyclic group satisfying the relations is trivial

Write the cyclic group as $$\langle g\rangle$$ and set

$$
x=g^{r},\qquad y=g^{s}.
$$

Insert these expressions in the defining relations:

$$
\begin{aligned}
xy^{2}=y^{3}x
    &\Longrightarrow g^{r+2s}=g^{3s+r},\\
yx^{2}=x^{3}y
    &\Longrightarrow g^{s+2r}=g^{3r+s}.
\end{aligned}
$$

In a cyclic group equality of elements means equality of the
↪   exponents
(modulo the order of $$g$$, or as integers if the order is
↪   infinite).  From the
first equality we obtain

$$
2s\equiv 3s\pmod{|g|}\;\;\Longrightarrow\;\;s\equiv0,
$$

and from the second equality

$$
2r\equiv 3r\pmod{|g|}\;\;\Longrightarrow\;\;r\equiv0.
$$

```
144  Thus $$r=s=0$$; consequently $$x=y=e$$.
145
146  -----------------------------------------------------------------⌋
     ↩   ----
147  ### 5.  Conclusion
148
149  The group defined by
150
151  $$
152  \langle x ,y\mid xy^{2}=y^{3}x ,\; yx^{2}=x^{3}y\rangle
153  $$
154
155  is cyclic, and the only cyclic group that satisfies the two
     ↩   relations is the
156  trivial group.  Hence
157
158  $$
159  \boxed{G\text{ is the trivial group}.}
160  $$
```

## B.18   GPT-5 THINKING

`Lines 12, 20, 35`: The definition of $[s,t]$ used is $sts^{-1}t^{-1}$. The commutator identity given then holds only if $s^t = tst^{-1}$ but the commutator definition is later taken to be $s^t = t^{-1}st$. (Critical)

```
Yes. Write the relations in commutator form and use a short
↪  lemma.

From

$$
xy^2=y^3x\quad\text{and}\quad yx^2=x^3y
$$

we get

$$
x y^2 x^{-1}=y^3\ \Longrightarrow\ [x,y^2]=x y^2 x^{-1}y^{-2}=y,
$$

$$
y x^2 y^{-1}=x^3\ \Longrightarrow\ [y,x^2]=y x^2 y^{-1}x^{-2}=x.
$$

Let $u=[x,y]$. Using the standard commutator identity
$[a,bc]=[a,b]\,[a,c]^b$, we have

$$
[x,y^2]=[x,y]\,[x,y]^y=u\,u^y,\qquad
[y,x^2]=[y,x]\,[y,x]^x=u^{-1}(u^{-1})^x.
$$

Hence

$$
y=u\,u^y,\qquad x=u^{-1}(u^{-1})^x. \tag{$\ast$}
$$

**Lemma.** If $a=w\,w^a$ in a group, then $a=w^2$.

*Proof.* $a=w\,w^a\Rightarrow 1=w\,w^a a^{-1}=w(a^{-1}wa)a^{-1}=w
↪  a^{-1} w$, so $a^{-1}=w^{-2}$ and thus $a=w^{2}$. $\square$

Apply the lemma to $(a,w)=(y,u)$ and $(a,w)=(x,u^{-1})$ in
↪  $(\ast)$:

$$
y=u^2,\qquad x=(u^{-1})^2=u^{-2}.
$$

Thus $x=y^{-1}$.

Substitute $x=y^{-1}$ into $xy^2=y^3x$:

$$
```

```
48  y^{-1}y^2=y^3y^{-1}\ \Longrightarrow\ y=y^2 \ \Longrightarrow\
    ↪   y=1,
49  $$
50
51  whence $x=1$ as well. Therefore $G$ is the trivial group.
    ↪   $\square$
```

