# OpenReview forum: "No LLM Solved Yu Tsumura's 554th Problem"
_ICLR.cc/2026/Conference — ICLR 2026 Conference Withdrawn Submission_

### Official Review · Reviewer_bR2g · 2025-10-14

**Soundness:** 3
**Presentation:** 1
**Contribution:** 1
**Rating:** 2
**Confidence:** 4

**Summary:**

The authors show that LLMs cannot solve Yu Tsumura’s 554th problem, despite the problem being below the difficulty of an IMO problem and its solution being available online. Additionally, they make one IMO participant solve the problem, showing that it falls within IMO difficulty level. Using their analysis, the authors argue for more caution of LLM's performances compared to the great buzz that came about due to the gold-level performance of some LLMs on the IMO 2025.

**Strengths:**

- Yu Tsumura's problem is a nice problem that does satisfy the constraints outlined in the abstract.
- Some counterargument against the buzz around IMO gold is appreciated.

**Weaknesses:**

I have split my concerns in three separate categories: a critical weakness, major weaknesses, and minor weaknesses.

**Critical Weakness**
The contribution of the work is basically non-existent. The authors find a **single** problem that LLMs cannot solve. It is very unsurprising that such a problem can be found, since even existing benchmarks also still contain such problems. The analysis is very limited, apart from showing that the models cannot solve the problem. In particular, it is never discussed how difficult it is to find such a problem, and no in-depth discussion is created about what this tells us exactly. A single problem cannot be used as a benchmark, cannot be used to do more quantitative analyses, and even qualitative analyses are limited due to the very specific nature of it being a single problem with a single model run. No interesting conclusions can be made based on this, apart from the fact that such a problem exists. Overall, this gives me more the impression that this could be an interesting blog post, but not a submission at a top-tier conference.

**Major Weaknesses**
- The critical weakness is further exemplified by the lack of a "Method" and "Related Work" section. "Method" is missing because there is no method that is presented by the authors, and "Related Work" should be much more thoroughly discussed in a separate section (especially because the authors have plenty of space left).
- Each model is only evaluated once on the problem. While the authors argue that this simulates end-user experience, it does not do so for every user: if you run it five times and it get the solution once, then 1/5 users will get the correct answer. For a full analysis of a model's performance, especially because it is only a single problem and evaluation is therefore quite noisy, one should evaluate a single model multiple times.
- It is unclear why the authors want to simulate the end-user experience, as the paper's point would become much stronger if (1) each model fails the problem using multiple attempts, (2) end-user experience is much more interactive especially for these problems, where end-users can decide to run multiple times or point out a mistake to the model. No simulation of this end-user experience is performed though.
- The reasoning to do only a single pass without any agentic framework (except for Gemini Deep Think) is shaky: the authors specifically mention that they wish to counteract the optimism that existed after the IMO gold medals, but these medals were only obtained by very heavy agentic frameworks that reasoned for hours, not minutes. As shown in the MathArena evaluation cited by the authors, even a best-of-32 method does not get bronze, clearly indicating that the models evaluated by the authors are not even capable yet to perform on IMO-level problems.
- The IMO participant clearly indicates in their proof-writing that they spent well over 5h on the problem. This is much more than what one could spend on a single problem in the actual IMO and somewhat contradicts the author's claims that the problem should be solvable relatively easily by systems that perform well on the IMO (L61-65, L193-195). This point is never discussed, but if anything, it shows that the problem is in fact rather tough and definitely **not** well within the capabilities of IMO-level participants (as claimed in L193-195). This makes the fact that models do not find the solution even less surprising.
- The authors claim in the abstract that they show differences in proof quality apart from proof correctness (by looking at "motivated proofs"). This claim is only substantiated on a single human-written proof, on only a single problem that the LLMs struggle with. Therefore, it is not substantiated at all. To substantiate this claim more rigorously, the authors should evaluate models on a wider range of problems, specifically on ones that LLMs can also solve to compare proof quality between correct proofs. No such analysis is made, and therefore the claim cannot be made.

**Minor Weaknesses**
- The writing in the paper is quite informal and would benefit from several more passes to improve writing and structure. This point is exemplified by the lack of some sections (as mentioned above) and the structure in section 3: very short paragraphs, commenting that something is "much nicer", "until the whole thing collapses", " etc.)
- There are various typos etc. throughout the work, indicating a lack of rigor in the write-up that is not acceptable for a top-tier conference submission. For instance,
- - L27: remove the "."
- - L40-42: natural-language -> natural language, system -> systems
- - L86: "different" missing
- - L93-94: "making ... should be" is not a proper sentence.
- - L100: website -> websites
- - L145: "a" missing
- - L181: "at" -> "in"
- - L191: "form" -> "from"
- - L206: "participants" -> "participant"
- - L236: "genereted"
- The claim in L20 that the proof by the IMO participant is better than the publicly-available proof is never discussed and irrelevant for the paper.
- The comment in L343 is a bit strange, as it does not list the most likely reason for the discrepancy: different model runs lead to different reasoning, which could have caused the model to reason for longer in one attempt than in the other. If the author's would have used the API, they could have checked this by storing the number of tokens the model used.

**Questions:**

See above

---

### Official Review · Reviewer_bqyU · 2025-10-15

**Soundness:** 1
**Presentation:** 1
**Contribution:** 1
**Rating:** 0
**Confidence:** 5

**Summary:**

The paper argues that despite recent high-profile successes of LLMs on Olympiad-style mathematics, there exists at least one publicly available problem—Yu Tsumura’s 554th group-theory problem—that no off-the-shelf LLM (proprietary or open-weight) reliably solves. The authors evaluate 16 contemporary models in a one-shot setting, annotate failure modes (algebraic errors, missing cases, inapplicable theorems, etc.), and claim all models fail critically. They also include a small human comparison: a former IMO participant (without prior group-theory background) solves the problem and proposes a more “motivated” proof than the canonical solution. The paper concludes that current LLM reasoning remains brittle and that proof-based evaluation (beyond final-answer scoring) is needed.

**Strengths:**

N.A.

**Weaknesses:**

- **Extremely narrow empirical scope:** The central claim rests on one hand-picked problem. Even if illustrative, this is a qualitative case study rather than a robust empirical evaluation; conclusions about “systematic failure” risk overgeneralization.
- **Human comparison is anecdotal (n=1):** The IMO participant case study is interesting but not a controlled experiment; it cannot support claims about why humans succeed where LLMs fail, nor establish that the task is broadly “within IMO reach.”
- **Limited actionable insight:** The paper hypothesizes causes (search depth, algebraic error accumulation) but does not translate them into concrete diagnostics or interventions (e.g., ablations, controlled perturbations, or tool-use tests) that practitioners can adopt.
- **Minimal open-source contribution:** Beyond posting traces and a transcript, there is no released evaluation harness, reproducible prompts/configs, or a curated suite of related problems to make this a reusable benchmark to benefit the open-source community.
- **Causal claims vs. evidence:** The manuscript suggests that proof-based reasoning is uniquely lacking relative to final-answer tasks, but with only one problem it’s hard to disentangle: prompt sensitivity, contamination, internal sampling policies, and tool-use constraints. Moreover, Formal AI4Math [1] is just designed to overcome this kind of proof case. The authors apparently lack enough effort in the literature review.

In general, this is not a complete ICLR submission. There are many directions that the authors could dive deeper into.

[1] Yang, Kaiyu, et al. "Formal mathematical reasoning: A new frontier in ai." arXiv preprint arXiv:2412.16075 (2024).

**Questions:**

N.A.

---

### Official Review · Reviewer_CZVY · 2025-10-28

**Soundness:** 2
**Presentation:** 2
**Contribution:** 2
**Rating:** 0
**Confidence:** 4

**Summary:**

This paper focuses on evaluating the abilities of several LLMs to prove the Yu Tsumura’s 554th problem in natural language. The paper reports that 18 LLMs all fail to prove this problem in natural language. It further defines 6 types of errors and labels each LLM output with one or more of those labels trying to provide insights about what goes wrong.

**Strengths:**

The topic is interesting. It is always insightful to study and investigate the shortcomings of existing models. This may eventually lead to improving the models.

**Weaknesses:**

Contribution of the paper is very narrow and limited in my view. I think the paper reads more like a blog post rather than a technical paper ready to be peer reviewed. I am not sure what authors expect from this review process. If the paper believes it has made a significant contribution, this might imply lack of familiarity with the literature and the CFP of ICLR.

The experiments are weak only focusing on a single problem. The investigation is insightful, but limited. This investigation could have been performed on an entire set of problems, i.e., a whole dataset. After such an analysis, it would be useful for a reader to see experiments about how an AI system can gain the capability to solve such problems. Merely reporting that some models cannot solve/prove a single problem is not a contribution significant enough to publish a paper.

Some of the claims in the paper appear weak and unfounded to me. For example, I do not agree with the claim that Yu Tsumura’s 554th problem is similar to IMO problems. The underlying topic, the level of difficulty, etc are significantly different. If the paper provides empirical evidence supporting their claim, I would not disagree with it. For example, if the paper surveys high school students preparing for IMO with no education beyond the high school to prove this problem and such students achieve similar performance as they get on IMO, that would be convincing. In the absence of such evidence, I do not agree with those claims.

Group theory has its own language and notation. When a person or a LLM is not educated on the language of group theory, they will not be able to prove problems related to group theory. Paper does not make an effort to identify the underlying problems that might prevent people and LLMs in proving the Yu Tsumura’s 554th problem. For example, the paper can investigate how familiar LLMs are with the basics of group theory. Are there problems easier than the 554th problem that some LLMs can prove easily?

The starting point and the claims of the paper seem to be in response to people's optimism about the mathematical capabilities of LLMs. However, such optimism is not necessarily present in the literature. Therefore, in my view, the paper remains disconnected from the published results in the literature.

The models that paper investigates are different from the models that were able to reach the level of gold medal at IMO.

The paper can consider decomposing the proof into its building blocks and further analyze the capabilities of LLMs in proving those building blocks. The paper can distinguish between the ability of a model to prove individual steps in the proof and the ability to plan a proof, i.e., put the individual pieces of proof together and then prove the theorem. This could have been some insightful results.

Despite the fact that paper only studied one single problem, it does not make the effort to study the capabilities of LLMs in proving the problem in a formal language such as Lean.

**Questions:**

Are authors aware that the models they have used in their experiments are different than the models that have reached the level of gold medal at IMO?

---

### Official Review · Reviewer_fFT5 · 2025-11-02

**Soundness:** 2
**Presentation:** 3
**Contribution:** 1
**Rating:** 2
**Confidence:** 4

**Summary:**

This paper presents a counterclaim to the optimism about LLM's problem-solving abilities, which was fueled by recent gold medals at the International Math Olympiad (IMO). The authors show that a problem exists—Yu Tsumura's 554th problem—that is within the scope of an IMO problem, is not a combinatorics problem, requires fewer proof techniques than typical hard IMO problems, has a publicly available solution, and cannot be readily solved by any existing off-the-shelf LLM. The paper includes an analysis of the output traces of 16 SOTA LLMs. Additionally, the authors compare the generic LLM output to a new proof by a former IMO participant. This comparison, which is to a proof that is significantly better motivated than the original, is used to elaborate on the differences in LLM and human proof quality.

**Strengths:**

1. This paper identifies an interesting phenomenon where existing SOTA LLMs fail to solve a math problem that has a publicly available solution and is of moderate difficulty.
2. The paper's writing is clear, and its argument is explicit, making the authors' central claim easy to understand.

**Weaknesses:**

1. The paper does not provide many reliable directions for this identified problem, such as improved training data or experiments.
2. The core contribution of the paper relies entirely on a single data point (one problem), which limits the generality of the conclusion about LLM reasoning "brittleness."
3. The authors acknowledge this specific problem will likely be "patched" by models according to Goodhart's Law, which limits the long-term contribution value of this specific finding.

**Questions:**

1. The evaluation was conducted one-shot. Would the results have been different if more complex prompting strategies (like CoT, self-reflection) or multiple sampling (pass@k) were used?
2. The authors assume the problem's solution is "likely" in the training data; is there a method to verify this more concretely? If it is not in the training data, wouldn't this just be a normal, hard problem?

---

### Note · Authors · 2026-01-23

**Comment:**

We thank the reviewers for their feedback and are withdrawing the paper.

Our read of the main critique is that it focuses on a “single data point” (all reviewers).

Regarding this point, we want to emphasize that this problem was not cherry-picked (reviewer bqyU): it emerged at the end of a months-long, exhaustive search. We started from 300+ group theory theorems. Based on this prior work, another list of 22 problems was formulated, which were put through several LLMs. We did not observe a clear pattern in this case of one problem consistently defeating all LLMs. Based on this list and prior experiments, we arrived at Yu Tsumura's 554th problem. Because the prior work on which the discovery of Yu Tsumura's 554th problem rested, that defeated all LLMs at the time, was not done in a fully rigorous manner, we did not include the description of this process in the paper. Based on the review we received, it is clear that this process should have been described in greater detail. One other point to note here, although we didn't mention this, is that even when given a wordy description of some of the key features of the correct solution, no LLM could leverage this information and find a solution. It is also noteworthy that LLMs are able to solve much harder problems that Yu Tsumura's 554th problem, indicating a "non-transitivity" compared to human problem solving, where solving a harder problem implies the ability to solve an easier one.

We agree with the referees that the paper does not identify reliable directions to take this work. We could have tried to speculate on reasons that LLMs make the kinds of errors we classified, but we preferred to keep the narrative of the paper focused.

We  disagree with the suggestion that the “typical user environment” should be modeled via pass@k plus CoT or self-reflection strategies In practice, mathematicians and students do not run elementary problems dozens of times with a harness and then adjudicate outputs with other models, and our goal was to analyzing a single shot user-interaction mode.  This indicates a difference of point of view from ourselves and some of the reviewers. Further, pass@k is only realistically possible with autograding capabilities. We believed on the other hand human grading was necessary for reliability—and the effort required to produce publishable, human-verified evaluations was already at (and beyond) what we could complete for such a project.

There are some minor comments that surprised:
-  e.g. reviewer bR2g mentioned that our comparison was not fair because the we did not normalize for compute time: *"IMO gold medals [...] were only obtained by very heavy agentic frameworks that reasoned for hours, not minutes. [...] The IMO participant clearly indicates in their proof-writing that they spent well over 5h on the problem. This is much more than what one could spend on a single problem in the actual IMO and somewhat contradicts the author's claims"*.
On the first point, we believe that unless one has a notion of token throughput, one cannot argue in this way, as even fast models might run on highly parallelized hardware in a short amount of time, consuming many tokens. On the second point, it is true that the high school student we surveyed did take more than five hours, whereas IMO problems are solved in less time. However, we don't believe this undercuts our claim. High school students do not receive elite training in group theory. We felt that the important question is whether a high school student could solve the problem unassisted, i.e. that essentially no specialised expertise needs to be acquired.
- e.g. reviewer CZVY mentioned as a weakness "For example, the paper can investigate how familiar LLMs are with the basics of group theory. Are there problems easier than the 554th problem that some LLMs can prove easily?" We think that it is completely obvious that modern LLMs understand the rudiments of group theory (and much more);

We conclude by noting that whilst we believe that the referees underestimated the amount of work required to identify a problem that fulfilled all the criteria of the study (able to be solved by a high school student, uniformly defeats all SOTA LLMs, not combinatorics, with a public solution and models that are publicly available), and evaluate this problem with respect to all leading SOTA LLMs of the time, we agree that more work is needed and wish to withdraw to conduct that work.

Since publicising our work, some modern LLMs can more or less reliably solve this problem, indicating that our work was not entirely in vain.

**Withdrawal Confirmation:**

I have read and agree with the venue's withdrawal policy on behalf of myself and my co-authors.